# Allometric equation for *Raphia laurentii* De Wild, the commonest palm in the central Congo peatlands

**Yannick Enock Bocko**[1]*, **Grace Jopaul Loubota Panzou**[1,2], **Greta Christina Dargie**[3], **Yeto Emmanuel Wenina Mampouya**[4], **Mackline Mbemba**[4], **Jean Joël Loumeto**[1], **Simon L. Lewis**[3,5]

**1** Laboratoire de Biodiversité, de Gestion des Ecosystèmes et de l'Environnement, Faculté des Sciences et Techniques, Université Marien Ngouabi, Brazzaville, République du Congo, **2** Institut Supérieur des Sciences Géographiques, Environnementales et de l'Aménagement (ISSGEA), Université Denis SASSOU-N'GUESSO, Kintélé, République du Congo, **3** School of Geography, University of Leeds, Leeds, United Kingdom, **4** Ecole Nationale Supérieure d'Agronomie et de Foresterie, Université Marien Ngouabi, Brazzaville, République du Congo, **5** Department of Geography, University College London, London, United Kingdom

* byannickenock@gmail.com

**Data Availability Statement:** The data used in this paper are freely available from the CongoPeat website: https://congopeat.net/data.

## Abstract

The world's largest tropical peatland lies in the central Congo Basin. *Raphia laurentii* De Wild, the most abundant palm in these peatlands, forms dominant to mono-dominant stands across approximately 45% of the peatland area. *R. laurentii* is a trunkless palm with fronds up to 20 m long. Owing to its morphology, there is currently no allometric equation which can be applied to *R. laurentii*. Therefore it is currently excluded from aboveground biomass (AGB) estimates for the Congo Basin peatlands. Here we develop allometric equations for *R. laurentii*, by destructively sampling 90 individuals in a peat swamp forest, in the Republic of the Congo. Prior to destructive sampling, stem base diameter, petiole mean diameter, the sum of petiole diameters, total palm height, and number of palm fronds were measured. After destructive sampling, each individual was separated into stem, sheath, petiole, rachis, and leaflet categories, then dried and weighed. We found that palm fronds represented at least 77% of the total AGB in *R. laurentii* and that the sum of petiole diameters was the best single predictor variable of AGB. The best overall allometric equation, however, combined the sum of petiole diameters ($SD_p$), total palm height ($H$), and tissue density ($TD$): $AGB = Exp(−2.691 + 1.425 × \ln(SD_p) + 0.695 × \ln(H) + 0.395 × \ln(TD))$. We applied one of our allometric equations to data from two nearby 1-hectare forest plots, one dominated by *R. laurentii*, where *R. laurentii* accounted for 41% of the total forest AGB (with hardwood tree AGB estimated using the Chave et al. 2014 allometric equation), and one dominated by hardwood species, where *R. laurentii* accounted for 8% of total AGB. Across the entire region we estimate that *R. laurentii* stores around 2 million tonnes of carbon aboveground. The inclusion of *R. laurentii* in AGB estimates, will drastically improve overall AGB, and therefore carbon stock estimates for the Congo Basin peatlands.

**Funding:** The following awards funded this study: a Natural Environment Research Council large grant (NE/R016860/1; http://gotw.nerc.ac.uk/list_full.asp?pcode=NE%2FR016860%2F1&cookieConsent=A) awarded to S.L.L., a British Ecological Society Ecologists in Africa (EA21/1263; https://www.britishecologicalsociety.org/funding/ecologists-in-africa/) grant awarded to Y.E.B The funders had no role in study design, data collection and analysis, decision to publish, or preparation of the manuscript.

**Competing interests:** The authors have declared that no competing interests exist.

## Introduction

Improved estimates of the carbon stocks and flows in tropical ecosystems is critical to quantifying carbon loss from land-use change, and carbon uptake in remaining ecosystems [1]. For example, the implementation of REDD+ (reducing emissions from deforestation and forest degradation) in tropical countries is crucial for climate change mitigation [2]. Each country engaged in the REDD+ process must provide information on their REDD+ strategies and carbon reference levels [3]. However, information on reference levels requires accurate estimates of the biomass, and therefore the carbon stock, of the different forest types.

For the countries of the Congo Basin establishing carbon reference levels is hindered by a lack of data. This is particularly true for swamp forest ecosystems. The peatlands of the Cuvette Centrale, Congo Basin, are estimated to cover 167,600 km$^2$ and store 29.0 Pg C belowground in the peat [4]. However, *in situ* data on aboveground biomass from these peatland ecosystems is rare [5–7]. Additionally, across approximately 45% of these peatlands the palm species *Raphia laurentii* De Wild [5] forms dominant or monodominant stands, but at present there is no allometric equation which can be applied to *R. laurentii* to estimate its aboveground biomass (AGB), owing to its morphology. *R. laurentii* is a cespitose palm (multiple shoots from the same root system) with 4 to 6 stems (i.e. short trunks) which grow to 2–7 m tall and up to 20 cm in diameter [8]. Their fronds, however, can reach twice that length. We refer to *R. laurentii* as a trunkless palm, despite the presence of a stem in mature individuals, because their growth in compact clumps and the high number of fronds per clump, means the stem is not accessible and rarely visible. This makes it difficult to measure the diameter at the base of the stem or at 1.30 m from the ground, without destructive sampling. As a result it is not practically possible to apply the pre-existing palm allometric equations [9] and therefore *R. laurentii* has been excluded from previous aboveground biomass estimates of central Congo peat swamp forest, leading to a systematic bias and underestimation of carbon stocks [5–7].

Globally, the contribution of palms to forest biomass and productivity has been a neglected area of study, and therefore a source of uncertainty in forest carbon stocks and dynamics [10]. Whilst some allometric equations have been developed for Amazonian [9, 11, 12] and Asian [13, 14] palm species, these palms are morphologically dissimilar to *R. laurentii*. Generally, in monocotyledonous species, using a species-specific model is better than using a multispecies model [9]. Therefore, the development of an allometric equation for what is probably the dominant canopy plant in the world's largest tropical peatland complex, *R. laurentii*, is a fundamental contribution to understanding the carbon cycle in the Congo Basin peatlands. Without this information to guide sustainable land management policies, the peatlands are at risk from inappropriate development, which could have serious consequences for the region [15].

By linking the AGB of a species to certain morphological variables that can be measured in the field [16], allometric equations permit the AGB of a species to be estimated non-destructively. The morphological variables that can be used to establish an allometric equation are numerous and differ depending on the species. Diameter at breast height (DBH), wood density, height [17] and crown diameter [18] are, in descending order, the most important morphological variables in the estimation of the biomass of a dicotyledonous species [19, 20]. In contrast, for monocots, depending on the architectural type of the stem, the number of palm fronds, dry mass fraction, tissue density, DBH, diameter at the base of the stem, crown diameter, total height and stem height are often used as morphological predictor variables of AGB [9, 21, 22]. For palms in particular, height and diameter (at the base or at breast height) are the most important morphological variables for estimating above-ground biomass [9, 13], but for palms with morphologies like *R. laurentii*, other predictor morphological variables may be more practical and therefore preferential.

The aim of this study is to develop an allometric equation for the monocotyledonous species *R. laurentii*, in order to improve the aboveground biomass and carbon stock estimates of the peat swamp forests in the Congo Basin.

## Material and methods

### Study area

Fieldwork for this study was carried out in February 2019 in the Likouala Department, Republic of the Congo. The study site is located near to the village of Bolembe (1.190288N and 17.84239E; Fig 1) in a peat swamp forest which occupies an interfluvial basin, extending ~60 km between the black-water Likouala-aux-herbes River and the white-water Ubangui River, with an elevation of around 310 m a.s.l. [23]. The mean annual temperature and precipitation are, respectively, 25.6°C and 1556.5 mm [24]. The peatland is shallowly domed [23] and the vegetation is composed of two communities; one dominated by dicotyledons such as *Carapa procera* DC., *Symphonia globulifera* L. f., *Uapaca mole* Pax, *Grosera macranta* Pax, *Manilkara fouillonna* Aubrev. & Pellegr. and *Drypetes occidentalis* (Mull. Arg.) Hutch., termed hardwood-dominated peat swamp vegetation; the other by monocotyledons such as *Raphia laurentii*, *Aframomum angustifolium* (Sonn.) K. Schum. and *Pondanus candelabrum* P. Beauv., termed palm-dominated peat swamp vegetation [5, 25, 26], with a gradation between these

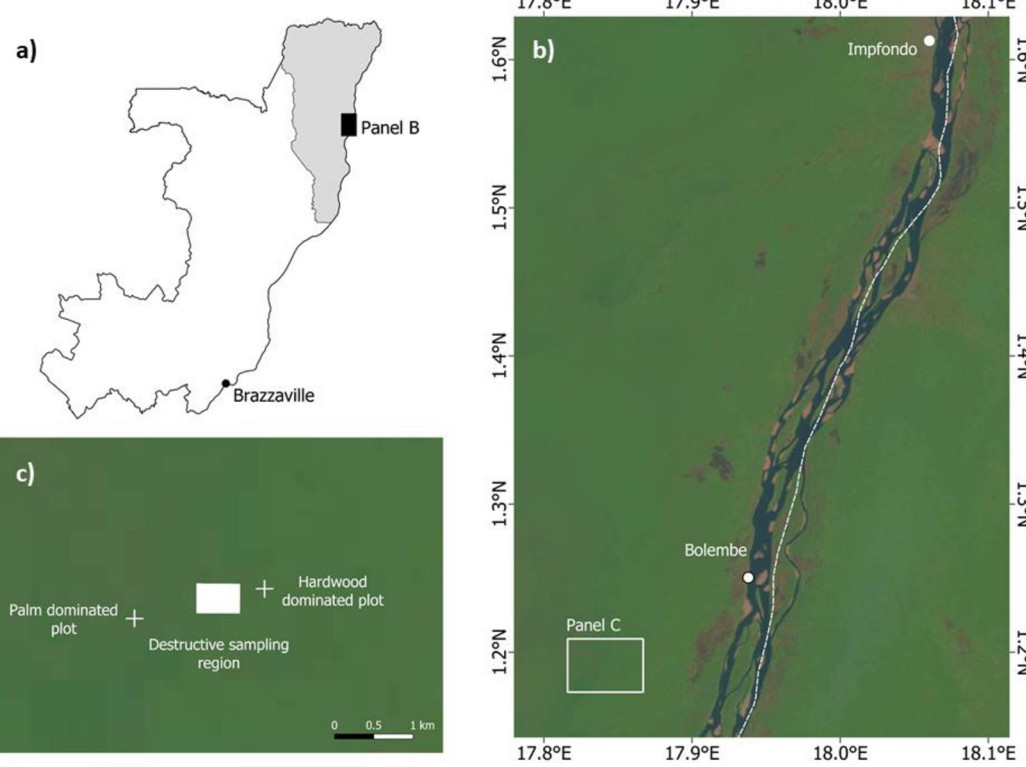

**Fig 1.** a) The location of the region of study (black solid rectangle, representing panel b) within the Republic of the Congo. The Likouala Department is highlighted in grey. b) The location of the fieldsite (white outlined rectangle, representing panel c) in relation to Bolembe village and Impfondo, the capital of the Likouala Department. The white dashed line represents the international border with the Democratic Republic of the Congo. The background is a Landsat 8 OLI/TIRS image (LC08_L1TP_181059_20200215_20200823_02_T1_refl; available from https://earthexplorer.usgs.gov/) c) The relative positions of the palm dominated and hardwood dominated swamp forest plots (white crosses) and the region where the 90 *R. laurentii* individuals were destructively sampled (white solid rectangle) displayed over the Landsat 8 OLI/TIRS image.

two vegetation types often spanning many kilometres. The fieldwork was conducted in a palm-rich area of swamp forest overlying ~2 m of peat (Fig 1). A field permit (No. 078/UMNG/VR-RC/DR) was obtained from the Vice-Recteur, head of Research and Cooperation, at the University of Marien Ngouabi.

## Palm selection and sampling

In order to construct a robust allometric equation which could be applied to the full size-range of *R. laurentii*, palm frond diameter data were collected prior to destructive sampling from a nearby 1 ha forest plot (EKG-03; 1.191998N, 17.84693E), located in a *R. laurentii* dominated forest. The diameter of each palm frond petiole was measured using callipers at 1.30 m from the ground where possible or, when it was inaccessible at this point, slightly above 1.30 m. Petiole diameter varied between 2 and 11.5 cm, both between and within *R. laurentii* individuals. Based on the average petiole diameter for each *R. laurentii* individual in the plot, we chose the following six mean petiole diameter classes for our destructive sampling: 2 to 4 cm, >4 to 5 cm, >5 to 6 cm, >6 to 7 cm, >7 to 8 cm and >8 cm. Destructive sampling was carried out approximately 1 kilometres away from the forest plot in an area around 400 m by 500 m. For each mean petiole diameter class we selected 15 individuals at random, each with at least three palm fronds, to be destructively sampled, giving a total of 90 stems.

Prior to felling, the number of live palm fronds was counted [27] and the diameter at the base of the stem was taken approximately 13 cm from the ground [13]. The total height, defined as the length from the base of the stem to the top of the fronds, i.e. the highest point above the ground [11], was measured using a hypsometer (manufacturer: Haglöf Sweden, Långsele, Sweden; model: Vertex IV). This measure of height, rather than for example stem or frond length, was taken because it can be measured non-destructively in the field. The diameter of each petiole was measured using callipers at 1.3 m from the ground or, depending on the point of petiole emergence, above 1.3 m, whilst avoiding the sheath.

## Destructive sampling and biomass data

Destructive sampling was carried out using a chainsaw. Each individual was cut down at the base, before separating each palm frond from the stem via the chainsaw at the base of the sheath. Each individual was then divided into stem (including the base of the sheath) and palm frond [9, 28] components. Each palm frond was further subdivided into four parts: sheath, petiole, rachis and leaflets. The leaflets were detached from the rachis using a machete. The fresh mass of each palm frond part was weighed in the field using a hand-held balance (manufacturer: PCE Instruments, Southampton, UK; model: PCE-CS, 300 kg capacity) and the lengths of the sheath, petiole and rachis were measured. From each *R. laurentii* individual, three palm fronds were selected and from these fronds five 15 cm samples [14] were collected from the middle of the sheath, the middle of the petiole, at the base of the rachis, the middle of the rachis and the top of the rachis. A single sample was taken from the leaflets (a minimum of 400 g), giving a total of 18 samples per *R. laurentii* individual. For the stem, the length was measured, before we cut it into sections. A disc of 5 cm was collected from the base, the middle and the top of each stem and then weighed with a 1 or 5 kg capacity balance (manufacturer: Kern, Balingen, Germany; model: Spring balance 281 and 285, respectively) before being sealed in plastic and transported to the laboratory.

In the laboratory, all samples from the stem and palm fronds were weighed using a suspended 6.2 kg capacity precision balance (manufacturer: Kern, Balingen, Germany; model: PLS 6200-2A, with a 0.01 kg precision). The volume of each stem, sheath, petiole and rachis sample was measured by hydrostatic weighing [20]. All samples were air dried before being

dried at 65˚C (for leaflets) and 101˚C (for the stem, sheath, petiole and rachis) [9] to constant mass [20].

The tissue density (*TD*) of each sample (stem, sheath, petiole and rachis) was calculated as dry mass divided by volume [29]. The mean dry mass/fresh mass ratio of each sample of the stem, sheaths, petioles, rachis and leaflets was used to convert the fresh mass weighed in the field, to dry mass. The total biomass of each *R. laurentii* individual was calculated by summing the dry mass of the stem, sheaths, petioles, rachis and the leaflets.

## Development of allometric equations

We developed a series of allometric equations using either single, two or three predictor variables first for *R. laurentii* individuals with a petiole mean diameter $\geq$2 cm and then for individuals with a petiole mean diameter $\geq$5 cm. This size division was to assess whether models which excluded smaller, juvenile *R. laurentii* individuals performed better than those which did not. The predictor variables used in the equations were stem base diameter (*D*), petiole mean diameter (*MD_p*), the sum of petiole diameters (*SD_p*), total palm height (*H*), tissue density (*TD*) and number of palm fronds (*N_pf*). To correct for heteroscedasticity, the predictor variables were log transformed. The single predictor variable linear models took the form of:

$$\ln(AGB_{est}) = a + b \times \ln(P_1)$$

Where $AGB_{est}$ is the aboveground biomass estimate for the *R. laurentii* individual, *a* is the intercept, *b* is the slope and $P_1$ is the predictor variable. The best single predictor variable model was selected according to the relative standard error (RSE), Akaike information criteria (AIC) and the Coefficient of Determination ($R^2$) value and was used as the basis for the two predictor variable models. For the two and three predictor variable models, predictor variables were first grouped into three types of parameters: a coarseness parameter (*D*, *MD_p* and *SD_p*), a height parameter (*H*) and the parameters intrinsic to each individual (*TD* and *N_pf*). For the two predictor variable models, the best single predictor variable was combined with one other predictor variable belonging to another parameter type (for example: *D* with *H*, *D* with *TD*, *MD_p* with *TD*, *SD_p* with *N_pf*). The two predictor variable model took the form of:

$$\ln(AGB_{est}) = a + b \times \ln(P_1) + c \times \ln(P_2)$$

Where $P_1$ is the best single predictor variable, *c* is a model parameter and $P_2$ is the second predictor variable. Again, the best two predictor variable model formed the basis for the three predictor variable model, which took the form of:

$$\ln(AGB_{est}) = a + b \times \ln(P_1) + c \times \ln(P_2) + d \times \ln(P_3)$$

Where in this instance $P_2$ is the best second predictor variable, *d* is a model parameter and $P_3$ is the third predictor variable.

The aboveground biomass estimated from each logarithmic linear model was multiplied by the following correction factor [30]:

$$CF = e^{(RSE^2/2)}$$

Where *CF* is the correction factor and *RSE* ($\sqrt{\frac{1}{n-p}\sum_{i=1}^{n}\varepsilon_i^2}$) is the residual standard error of the linear model. Where *n* is the sample size, *p* is the total number of predictor variables in the model and $\varepsilon$ is an error term.

The overall best performing model was the one with the highest $R^2_{adj}$ value, the smallest AIC, RSE, Roots Mean Square Error (RMSE) and Bias prediction

$(Bias\% = \frac{1}{n}\sum_{i=1}^{n}[(AGB_{est} - AGB_{observed})/AGB_{observed}])$ values [33]. A partial Fisher's test was used to test the validity of including the second and third predictor variables. All statistical analyses were carried with R (http://www.r-project.org/) [31], using the packages "ggplot2" [32], "minpack.lm" [33], "car" [34], "dunn.test" [35], "gvlma" [36] and "ggpubr" [37].

### Application of allometric equation

To assess the impact of the inclusion of *R. laurentii* on AGB estimates for the Congo Basin peatlands, we took data from two 1-ha forest peatland plots and compared the AGB estimates when palms were included and excluded. The two plots from which the data derived were the *R. laurentii* dominated plot (EKG_03), used to determine the mean petiole diameter classes, and a hardwood dominated swamp plot (EKG_02; 1.19200, 17.84693). Within these two plots the AGB of every tree with a DBH $\geq$10 cm was estimated using the allometric equation of Chave et al. [19]. As the only *R. laurentii* data available for these plots were the $N_{pf}$ for each individual, the $N_{pf}$ single predictor variable linear model for individuals with a mean petiole diameter $\geq$5 cm (m16, Table 2) was used to estimate *R. laurentii* AGB. We then compared the AGB estimates of the two plots when only tree AGB was included (as has previously been the practice [5]) and when both *R. laurentii* and tree AGB was included.

## Results

### Aboveground biomass partitioning

On average the 90 *R. laurentii* individuals destructively sampled had a height, from base to tallest frond, of 13.30 m, a stem base diameter of 21.67 cm, 5.7 fronds and a total AGB of 56.20 kg (Table 1). There was no significant difference in tissue density between the palm stem and petiole (p>0.05, Dunn's test; Table 1), but rachis and sheath tissue density were both significantly lower than the stem and petiole tissue density (p < 0.001, Dunn's test). The dry mass/fresh mass ratio was significantly higher (p < 0.001, Dunn's test) in the palm frond compartments, compared to the stem (Table 1).

Mean palm frond AGB (44.93±50.87 kg) was significantly higher than mean stem AGB (11.26±19.49 kg; p < 0.001, Kruskal-Wallis test; Fig 2a). AGB increased with increasing mean petiole diameter class (Fig 2b). The allocation of biomass between each component (i.e., stem, sheath, petiole, rachis and leaflets) was did not vary greatly between mean petiole diameter classes (Fig 3). Palm frond compartments accounted for more than 77% of the aboveground biomass in all mean petiole diameter classes (with the stem accounting for between 10–23% of the total biomass). On average the petiole accounted for the largest proportion of biomass (30.17%) followed by leaflets (25.17%), rachis (16.00%), stem (14.83%) and sheath (13.83%). While the proportion of rachis biomass increases with increasing mean petiole diameter classes, the proportion of leaflet biomass decreases (Fig 3).

### Allometric equations for AGB estimation

The sum of petiole diameters ($SD_p$) was found to be the best single predictor variable (m1, Table 2) for estimating *R. laurentii* AGB ($R^2$ = 0.97), followed by mean petiole diameter ($MD_p$), stem diameter ($D$), height from base to tallest frond ($H$) and number of palm fronds ($N_{pf}$) respectively, with all five variables showing a strong relationship with AGB (Table 2, Fig 4). Tissue density ($TD$) was the only single predictor variable to show a very weak relationship with AGB ($R^2$ = 0.07; Table 2, Fig 4).

**Table 1. Physical characteristics of the 90 *R. laurentii* individuals destructively sampled in the peat swamp forest, Likouala Department, Republic of the Congo.**

| Variable | | Mean (±St. Dev) | Range |
|---|---|---|---|
| Stem base diameter (*D*; cm) | | 21.67±6.96 | 8.20–35.40 |
| Petiole mean diameter (*MD$_p$*; cm) | | 6.01±2.11 | 2.22–12.54 |
| Sum of petiole diameters (*TD$_p$*; cm) | | 37.57±25.33 | 7.20–125.40 |
| Total palm height (*H*; m) | | 13.30±3.88 | 5.13–21.20 |
| Tissue density (*TD*; g cm$^{-3}$) | | 0.24±0.05 | 0.15–0.38 |
| No. of palm fronds (*N$_{pf}$*) | | 5.7±2.06 | 3–11 |
| Component Tissue density (g cm$^{-3}$) | Stem | 0.21±0.04 | 0.15–0.36 |
| | Sheath | 0.26±0.04 | 0.17–0.38 |
| | Petiole | 0.21±0.02 | 0.15–0.28 |
| | Rachis | 0.28±0.03 | 0.21–0.38 |
| | Leaflet | NA | NA |
| Component dry mass / fresh mass ratio | Stem | 0.22±0.04 | 0.16–0.35 |
| | Sheath | 0.30±0.04 | 0.20–0.40 |
| | Petiole | 0.42±0.04 | 0.30–0.55 |
| | Rachis | 0.48±0.03 | 0.39–0.55 |
| | Leaflet | 0.45±0.04 | 0.35–0.56 |
| Aboveground Biomass (kg) | Stem | 11.26±19.49 | 0.18–106.02 |
| | Sheath | 8.01±10.56 | 0.39–51.15 |
| | Petiole | 15.03±14.48 | 0.17–68.87 |
| | Rachis | 9.75±12.99 | 0.25–75.3 |
| | Leaflet | 12.14±13.99 | 0.67–89.98 |
| | Total | 56.20±69.13 | 2.03–386.34 |

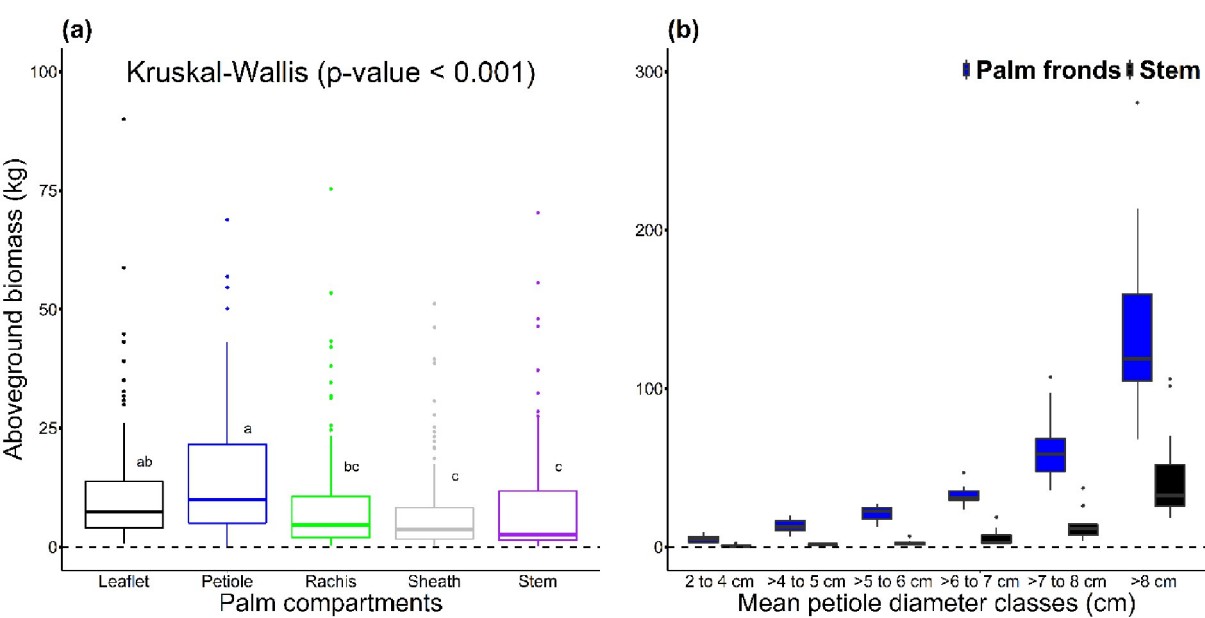

**Fig 2. Variability of aboveground biomass between leaflets, petiole, rachis, sheath and stem, and the results of a Kruskal-Wallis test (a) and between palm frond compartments collectively and stem across mean petiole diameter classes (b) for the 90 *Raphia laurentii* individuals destructively sampled.**

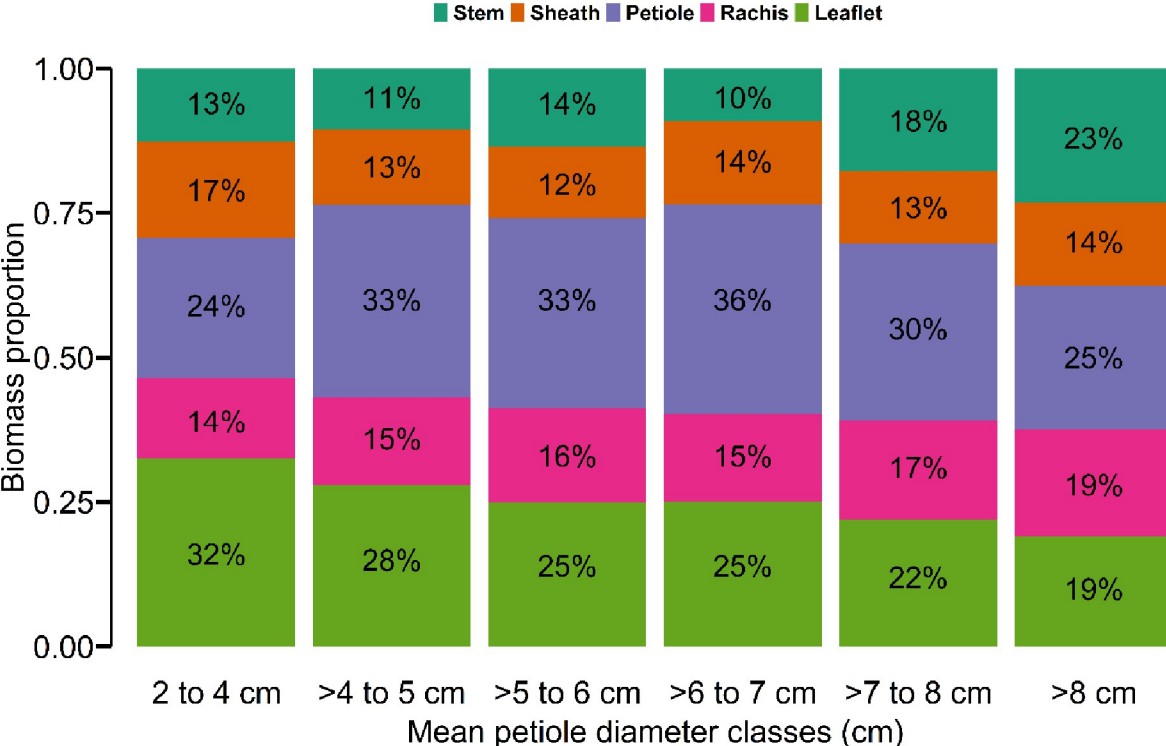

**Fig 3. Aboveground biomass distribution among the stem, sheath, petiole, rachis and leaflet for each mean petiole diameter class for the 90 *R. laurentii* individuals destructively sampled.**

When *H* was included as a second predictor variable (m7, Table 2), the model goodness fit of improved ($R^2$ = 0.98) and the *AIC, RSE, RMSE* and *Bias* values decreased (Table 2). However, the overall best performing model, with an $R^2$ = 0.98 and lowest the lowest *AIC* and RSE variables was a three predictor variable model which included $SD_p$, *H* and *TD* (m11, Table 2):

$$ln(AGB) = -2.691 + 1.425 \times ln\left(SD_p\right) + 0.695 \times ln(H) + 0.395 \times ln(TD)$$

When only individuals with mean palm petiole diameters greater than or equal to 5cm were used for model creation, the $SD_p$ was again the best single predictor variable ($R^2$ = 0.94, m12, Table 2), followed by $N_{pf}$, $MD_p$, *D* and *H*. *TD* was again the worst single predictor variable of AGB ($R^2$ = 0.94, m12, Table 2). The inclusion of *H* as a second predictor variable (m18, Table 2) gave the best performing model for this mean petiole diameter class ($R^2$ = 0.96 and lower RSE, AIC, RMSE and Bias values), but the addition of a third predictor variable did not improve model performance (Table 2). Both the best performing model overall (m11), and the best performing model developed for individuals with mean petiole diameters ≥5 cm (m18), underestimated AGB for larger *R. laurentii* individuals (Fig 5). Excluding smaller (i.e. <5 cm mean petiole diameters) only improved model performance when $N_{pf}$ was the single predictor variable (m5 vs. m16, Table 2).

When allometric model m16 was applied to the data from the two peat swamp forest plots, the contribution of *R. laurentii* individuals to the total AGB was 8% and 41% for the hardwood dominated (EKG-02) and *R. laurentii* dominated (EKG-03) peat swamp plots, respectively (Fig 6). The percentage difference between AGB estimates including and excluding *R. laurentii*

**Table 2. Allometric equations fitted to estimate *Raphia laurentii* aboveground biomass (*AGB*; kg dry mass) for two mean petiole diameter ranges.**

| Models | | Coefficient and exponent estimate | | | | Performance criteria | | | | | Correction Factor | Mean petiole diameter range (n) |
|---|---|---|---|---|---|---|---|---|---|---|---|---|
| N° | Formula | a | b | c | d | $R^2_{adj}$ | RSE | AIC | RMSE | Bias | | |
| **m1*** | $ln(AGB) = a + b \times ln(SD_p)$ | -2.496 | 1.727 | | | 0.97 | 0.21 | -19.48 | 15.44 | 0.02 | 1.02 | 2–15 cm (90) |
| **m2** | $ln(AGB) = a + b \times ln(D)$ | -6.308 | 3.211 | | | 0.86 | 0.44 | 113.53 | 39.28 | -4.58 | 1.10 | |
| **m3** | $ln(AGB) = a + b \times ln(MD_p)$ | -1.777 | 2.993 | | | 0.92 | 0.35 | 69.60 | 28.77 | -5.59 | 1.06 | |
| **m4** | $ln(AGB) = a + b \times ln(H)$ | -5.262 | 3.404 | | | 0.82 | 0.50 | 136.13 | 38.51 | -5.55 | 1.13 | |
| **m5** | $ln(AGB) = a + b \times ln(N_{pf})$ | -1.837 | 3.108 | | | 0.79 | 0.55 | 151.96 | 37.73 | 8.61 | 1.16 | |
| **m6** | $ln(AGB) = a + b \times ln(TD)$ | 7.792 | 3.206 | | | 0.07 | 1.15 | 285.26 | 66.88 | 8.78 | 1.94 | |
| **m7** | $ln(AGB) = a + b \times ln(SD_p) + c \times ln(H)$ | -3.318 | 1.431 | 0.721 | | 0.98 | 0.18 | -46.25 | 15.37 | -1.54 | 1.02 | |
| **m8** | $ln(AGB) = a + b \times ln(SD_p) + c \times ln(N_{pf})$ | -2.431 | 2.087 | -0.769 | | 0.97 | 0.19 | -37.86 | 15.97 | -2.01 | 1.02 | |
| **m9** | $ln(AGB) = a + b \times ln(SD_p) + c \times ln(TD)$ | -1.751 | 1.707 | 0.493 | | 0.97 | 0.21 | -23.09 | 15.93 | 0.22 | 1.02 | |
| **m10** | $ln(AGB) = a + b \times ln(SD_p) + c \times ln(H) + d \times ln(N_{pf})$ | -3.084 | 1.683 | 0.544 | -0.383 | 0.98 | 0.18 | -48.54 | 15.45 | -2.19 | 1.02 | |
| **m11**** | $ln(AGB) = a + b \times ln(SD_p) + c \times ln(H) + d \times ln(TD)$ | -2.691 | 1.425 | 0.695 | 0.395 | 0.98 | 0.18 | -49.15 | 15.71 | -1.55 | 1.02 | |
| **m12** | $ln(AGB) = a + b \times ln(TD_p)$ | -2.238 | 1.662 | | | 0.94 | 0.19 | -24.05 | 19.95 | -1.67 | 1.02 | 5–15 cm (60) |
| **m13** | $ln(AGB) = a + b \times ln(D)$ | -5.737 | 3.048 | | | 0.69 | 0.45 | 77.3 | 48.07 | -3.76 | 1.11 | |
| **m14** | $ln(AGB) = a + b \times ln(MD_f)$ | -2.573 | 3.400 | | | 0.77 | 0.39 | 59.84 | 33.89 | -5.59 | 1.08 | |
| **m15** | $ln(AGB) = a + b \times ln(H)$ | -5.027 | 3.349 | | | 0.64 | 0.48 | 86.29 | 45.59 | 0.27 | 1.12 | |
| **m16** | $ln(AGB) = a + b \times ln(N_{pf})$ | -0.272 | 2.353 | | | 0.80 | 0.36 | 50.73 | 41.86 | -2.59 | 1.07 | |
| **m17** | $ln(AGB) = a + b \times ln(TD)$ | 6.075 | 1.497 | | | 0.03 | 0.79 | 145.41 | 72.19 | -1.24 | 1.37 | |
| **m18** | $ln(AGB) = a + b \times ln(SD_p) + c \times ln(H)$ | -3.370 | 1.44 | 0.729 | | 0.96 | 0.17 | -38.32 | 18.41 | -1.25 | 1.01 | |
| **m19** | $ln(AGB) = a + b \times ln(SD_p) + c \times ln(N_{pf})$ | -2.478 | 1.964 | -0.491 | | 0.95 | 0.19 | -26.25 | 18.69 | -1.16 | 1.02 | |
| **m20** | $ln(AGB) = a + b \times ln(SD_p) + c \times ln(TD)$ | -1.920 | 1.653 | 0.209 | | 0.94 | 0.19 | -22.89 | 19.99 | -1.56 | 1.02 | |
| **m21** | $ln(AGB) = a + b \times ln(SD_p) + c \times ln(H) + d \times ln(N_{pf})$ | -3.375 | 1.559 | 0.678 | -0.169 | 0.95 | 0.17 | -36.87 | 18.06 | -1.43 | 1.01 | |
| **m22** | $ln(AGB) = a + b \times ln(SD_p) + c \times ln(H) + d \times ln(TD)$ | -3.232 | 1.440 | 0.718 | 0.080 | 0.95 | 0.17 | -36.00 | 18.44 | -1.24 | 1.01 | |

Predictor variables used in the equations are total palm height (*H*; m), stem base diameter (*D*; cm), total petiole diameter (*TD_p*; cm), mean petiole diameter (*MD_p*; cm), number of palm fronds (*N_pf*) and tissue density (*TD*; g cm$^{-3}$). Single asterisk (*) indicates the best performing single predictor variable model and double asterisks (**) indicates the best performing multiple predictor variable model.

was 9% and 71%, for the hardwood dominated and *R. laurentii* dominated plots, respectively (Fig 6).

## Discussion

Given that palm dominant or monodominant forests are much more likely occur in wetland ecosystems [10, 38], a lack of appropriate allometries for palms will have a larger impact on the accuracy of AGB and carbon stock estimations for tropical wetland ecosystems, such as the central Congo Basin peatlands studied here. This is because in areas with higher water tables, the shallow rooting systems of palms means they can out-compete deeper rooted dicots [10], resulting in a disproportionally high number of palms species with a tendency towards hyper-dominance [39]. For those ecosystems where tree palms (i.e. palms with a clearly defined trunk) dominate, a lack of a relationship between height and diameter in palms means the application of dicotyledonous allometric equations can lead to an over- or underestimation of AGB [9, 10]. As a result tree palm specific allometries have been developed [9]. Yet, in the

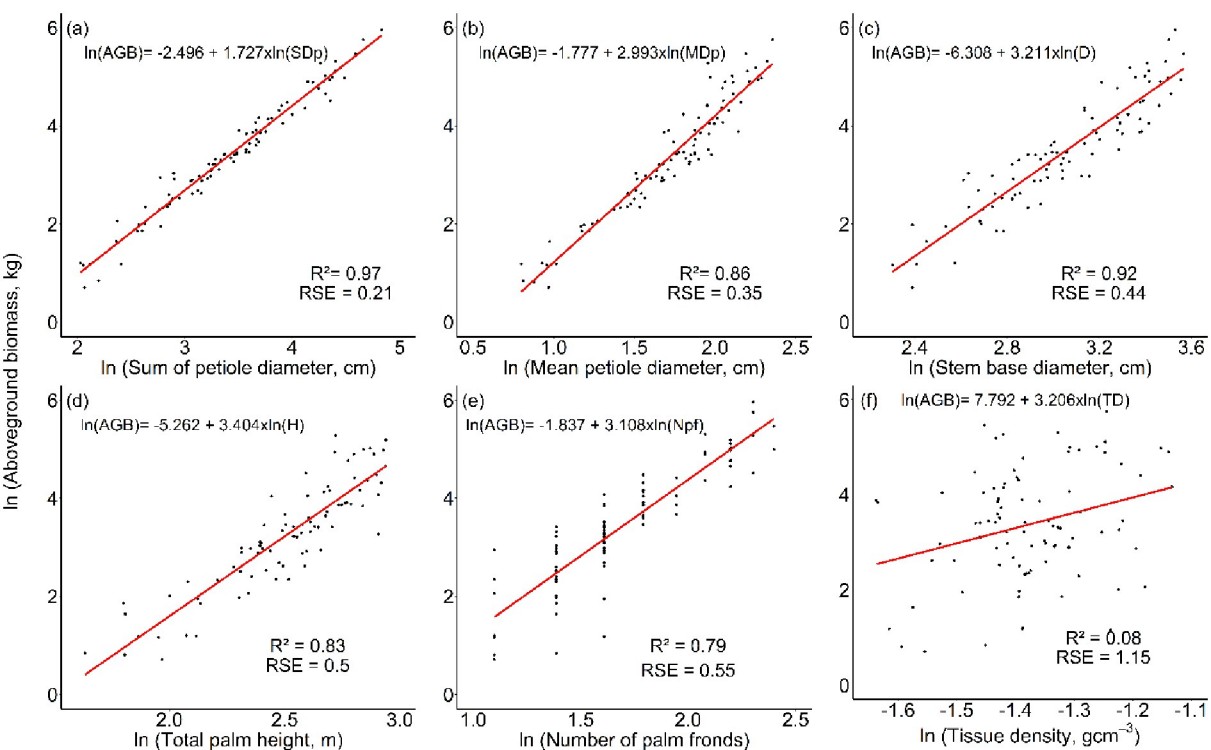

**Fig 4. Allometric relationships between the aboveground biomass of *R. laurentii* individuals and each single predictor variable.**

Congo Basin peatlands, the often dominant or mono-dominant *R. laurentii*, has a trunkless form, with allometry based on diameter not being applicable. Thus the contribution of AGB from *R. laurentii* has been excluded from previous assessments of the AGB of the peatlands of central Congo. Here provide the appropriate allometry for *R. laurentii*.

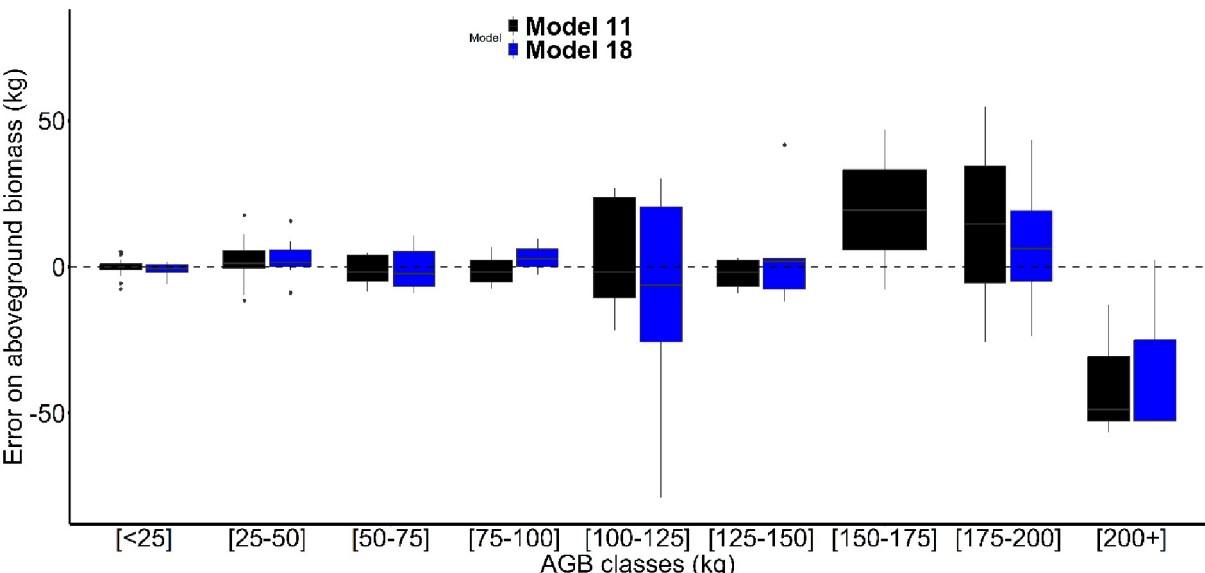

**Fig 5. Mean relative error for best performing aboveground biomass (AGB) models for *R. laurentii* individuals with mean palm petiole diameters of ≥2 cm (model 11) and ≥5 cm (model 18) across different AGB ranges.**

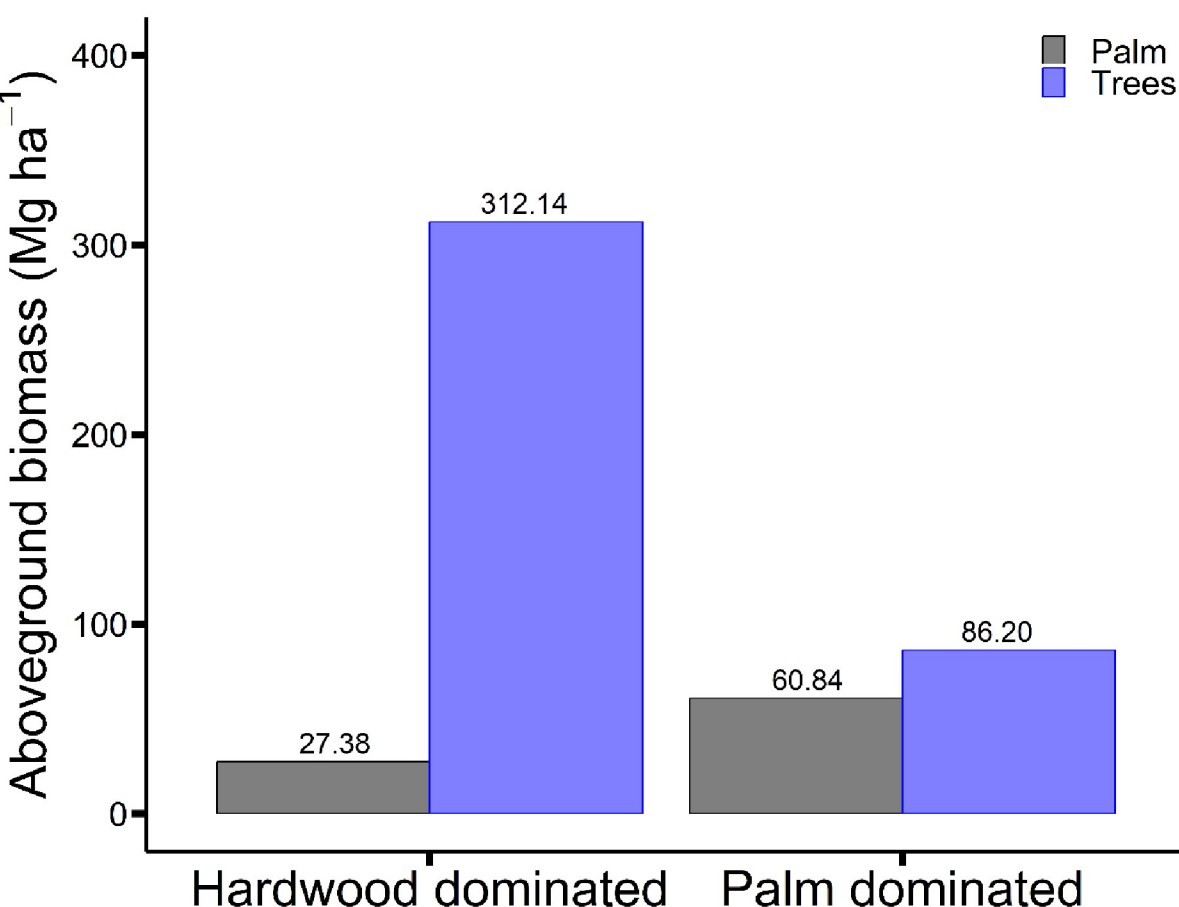

**Fig 6. The difference in AGB estimates for a hardwood dominated and *Raphia laurentii* dominated peat swamp forest plot when *R. laurentii* AGB estimates, calculated using allometric model m16, are included or excluded.**

### Aboveground biomass partitioning

The trunkless morphology of *R. laurentii* means it is not practical to adopt pre-existing allometries developed for other palm species, which rely on measurements of stem height or stem diameter [9, 11, 40]. Through our destructive sampling of *R. laurentii* individuals we show that the palm fronds account for the majority (77%) of their AGB. This is in contrast to similar studies of tree palm AGB for *Euterpe precatoria* Mart. in the Amazonian Basin [11] and *Elaeis guineensis* Jacq. in the Congo Basin [40], which found that the stem accounted for 81% and 73% respectively. Likewise Goodman et al. [9] found that stem height and to a lesser extent, stem diameter, were the best predictors of palm AGB across several Amazonian palm species. Here however, the concentration of AGB in the palm fronds, meant that stem diameter and total palm height were the some of the worst performing single variable predictors of *R. laurentii* AGB (Table 2). Furthermore, as we demonstrate, *R. laurentii* stems are not accessible without destructive sampling, meaning stem diameter based allometry is not possible. The vastly different architecture of *R. laurentii* means it requires its own unique allometric equation in order to estimate its AGB, which we provide.

### Allometric equation for AGB estimation

Overall the best model to estimate *R. laurentii* AGB was the three parameter model combining $SD_P$, *H* and *TD* (m11; Table 2). Although the use of tissue density in a species specific model

may initially seem redundant—as all individuals are from the same species with the same mean *TD*–we have kept this term as it is possible that tissue density may vary depending on environmental conditions. This may be the case give the central Congo peatlands cover 16.7 million hectares, thus *R. laurentii* are likely to be round growing under different environmental conditions, and may have systematically differing *TD*. *TD* can either be assumed to be the same as in this study or measured *in situ*. Alternatively, we also provide a three parameter model combining $SD_p$, *H* and $N_{pf}$ provides an almost equally good estimation of *R. laurentii* AGB, that does not include *TD*.

Single variable models, using either $SD_p$, $MD_p$, *D*, *H* or $N_{pf}$ also performed well when estimating *R. laurentii* AGB, with $SD_p$ as the best single variable to predict AGB (m1, Table 2). Unsurprisingly, *TD* was the only variable which was not suitable for predicting *R. laurentii* AGB in a single variable model, as it varies little among individuals within the study site. The single parameter models, particularly m16, which uses $N_{pf}$ as the predictor variable, have the advantage that they provide a good estimate of *R. laurentii* AGB with data which can be collected rapidly and at a low cost in the field.

## Regional palm AGB estimates

The significant contribution of *R. laurentii* to overall live AGB in the Congo Basin peatlands is evident when we compare the AGB estimates for the two peat swamp forest plots when *R. laurentii* is included and excluded. Despite the mean AGB of individual *R. laurentii* sampled being lower than that of the estimated AGB for individual trees measured, the high number of *R. laurentii* stems means this species accounted for 41% of AGB in the *R. laurentii* dominated forest plot. But even in the hardwood dominated forest plot, *R. laurentii* still accounted for 8% of the AGB. Thus, *R. laurentii* AGB is an important fraction of total AGB in the central Congo peatlands. The prevalence of *R. laurentii* across the Congo Basin peatlands means they are regionally an important component of forest AGB. Given that palm-rich peat swamp forest occupies an estimated 66,300 km$^2$ of peatland in the central Congo Basin [4], using our estimate of 30.4 Mg C ha$^{-1}$ for the palm-dominated plot, we estimate that approximately 2 million tonnes of carbon (i.e. 0.2 Pg C) is stored aboveground in *R. laurentii* across the region, compared with 1.4 Pg C estimated to be stored in the trees [5]. Combining our *R. laurentii* estimates with average tree aboveground carbon stocks for the region [5], we estimated that palm dominated peat swamp forest contains 97.4 Mg C ha$^{-1}$ and hardwood peat swamp forest contains 137.3 Mg C ha$^{-1}$.

## Conclusion

We have developed the first allometric equations for the palm species *Raphia laurentii*, the most abundant palm in the Congo Basin peatlands. We find that the majority of this trunkless palm's AGB is concentrated in the palm fronds (77%). Of the single predictor variables used in our allometric equations, the sum of petiole diameters was the best correlated variable used to estimate *R. laurentii* AGB. This is different to the findings of allometry studies of other palm species, which have shown stem height to be the key predictor variable of palm AGB. Overall the best estimates of *R. laurentii* AGB were achieved with a three parameter model which combined the sum of petiole diameters, height from base to tallest frond and tissue density. When we applied one of our allometric equations to data from two peat swamp forest plots whose AGB from trees ≥10 cm diameter was known, *R. laurentii* accounted for 41% of the total AGB in the palm dominated plot, and 8% in the hardwood dominated swamp plot. Over the entire region, we estimate *R. laurentii* accounts for ~2 million tonnes of C stored in its AGB, highlighting its importance to regional carbon stock, and the importance of developing allometries for common species with less common morphologies.

## Supporting information

**S1 Fig. Residual graphs of the model 11.**
(DOCX)

**S2 Fig. Residual graphs of the model 18.**
(DOCX)

**S1 Table. Tissue density values of the different compartments of an individual *Raphia laurentii.***
(DOCX)

## Acknowledgments

We thank the villages of Bolembe, Ekolongouma, Bethlem and Moungouma, for their assistance during the field data collection period. Thanks to the Ministry of the Environment, Sustainable Development and the Congo Basin, the Ministry of Forestry Economy and the Ministry of Higher Education, Scientific Research and Technological Innovation, Republic of the Congo, Marien Ngouabi University and the Likouala Department governor for the research permits. We thank the fieldworker team. We would also like to thank the University of Leeds (School of Geography) and the University of Edinburgh for providing hosting us during a scientific visit that enabled the finalisation of this manuscript. It is a pleasure to acknowledge Institut Nationale de Recherche Forestière (IRF) for the provision of a third oven for drying the samples. At the end of this work, we would also like to thank Suspense Ifo and Helen Plante for their logistical.

## Author Contributions

**Conceptualization:** Yannick Enock Bocko, Greta Christina Dargie, Simon L. Lewis.

**Data curation:** Yannick Enock Bocko, Grace Jopaul Loubota Panzou.

**Formal analysis:** Yannick Enock Bocko, Grace Jopaul Loubota Panzou.

**Funding acquisition:** Yannick Enock Bocko, Simon L. Lewis.

**Investigation:** Yannick Enock Bocko, Greta Christina Dargie, Yeto Emmanuel Wenina Mampouya, Mackline Mbemba, Simon L. Lewis.

**Methodology:** Yannick Enock Bocko, Grace Jopaul Loubota Panzou, Jean Joël Loumeto.

**Project administration:** Yannick Enock Bocko, Jean Joël Loumeto, Simon L. Lewis.

**Resources:** Jean Joël Loumeto, Simon L. Lewis.

**Software:** Yannick Enock Bocko, Grace Jopaul Loubota Panzou.

**Supervision:** Jean Joël Loumeto, Simon L. Lewis.

**Validation:** Yannick Enock Bocko, Grace Jopaul Loubota Panzou, Greta Christina Dargie, Yeto Emmanuel Wenina Mampouya, Mackline Mbemba, Jean Joël Loumeto, Simon L. Lewis.

**Visualization:** Yannick Enock Bocko, Grace Jopaul Loubota Panzou, Simon L. Lewis.

**Writing – original draft:** Yannick Enock Bocko.

**Writing – review & editing:** Yannick Enock Bocko, Grace Jopaul Loubota Panzou, Greta Christina Dargie, Yeto Emmanuel Wenina Mampouya, Mackline Mbemba, Jean Joël Loumeto, Simon L. Lewis.

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
