## [Decision Letter · Decision Letter 0]

4 Oct 2022

PONE-D-22-22408Allometric equation for the commonest palm in the Central Congo Peatlands, Raphia laurentii De Wild.PLOS ONE

Dear Dr. Bocko,

Thank you for submitting your manuscript to PLOS ONE. After careful consideration, we feel that it has merit but does not fully meet PLOS ONE’s publication criteria as it currently stands. Therefore, we invite you to submit a revised version of the manuscript that addresses the points raised during the review process.

We look forward to receiving your revised manuscript.

Kind regards,

John Toland Van Stan II, Ph.D.

Academic Editor

PLOS ONE

Journal Requirements:

5. We note that Figure 2 includes an image of a participant in the study. 

Additional Editor Comments:

Thanks for your submission! We have received reviews from three referees, two have uploaded a commented pdf. Please be sure to check these pdfs online. After reading the manuscript and these review reports, I agree that the manuscript requires major revision. Referees which suggest major revision focus on missing or unclear information in various sections (including methods and results/discussion). As such, please focus extra attention to the writing while addressing the reviewer concerns.

Reviewers' comments:

Reviewer's Responses to Questions

**Comments to the Author**

1. Is the manuscript technically sound, and do the data support the conclusions?

Reviewer #1: Partly

Reviewer #2: Yes

Reviewer #3: Yes

2. Has the statistical analysis been performed appropriately and rigorously? 

Reviewer #1: Yes

Reviewer #2: Yes

Reviewer #3: Yes

3. Have the authors made all data underlying the findings in their manuscript fully available?

Reviewer #1: Yes

Reviewer #2: Yes

Reviewer #3: Yes

4. Is the manuscript presented in an intelligible fashion and written in standard English?

Reviewer #1: No

Reviewer #2: Yes

Reviewer #3: Yes

5. Review Comments to the Author

Reviewer #1: The main goal of this manuscript was to develop allometric equations to estimate AGB for the palm R. laurentii, which dominates the peatlands of the Congo Basin, a critical area for carbon sequestration with the potential to influence global climate. The question is thus relevant, and the authors collected data from 90 individuals. Despite the importance of the question, the manuscript needs to improve significantly in the following aspects:

1. The authors need to significantly improve the use of the English language, the wording, and the paper organization. Many sentences are vague and unclear, have repeated terms, or similar words that follow one another, and the paper has repetitive sections. I tried to mark these issues in the manuscript, but it was a bit too much. The authors need to find help streamlining their writing.

2. The literature review needs to improve. The authors rely on revising allometric equations develop for palms of different growth forms relative to R. laurentii. Some of the comparisons are erroneous in that the compared palms have a significant DBH vs H relationship and R. laurentii is trunkless or has a short stem. Thus, the allometry of the target species is fundamentally different from that of published studies on palms of different growth forms. The discussion should go in the direction of improving the information for palms of different growth habits.

3. The use of the term “physical parameters” is confusing. Physical could refer to climatic conditions (temperature, radiation, precipitation, etc.) Use instead “morphological variables”.

4. Explain how the 90 palms were selected. Was it a random selection? How spread out were there at the study site? What were the criteria for selection?

5. The figures need to improve. Figure 1´s quality is low. Provide a regional map of the country indicating the study area. Panel b in figure 1 is very dark with a small font size that is hard to read. Figure 2 is unnecessary. The legend of all figures needs to improve and be more accurate. See my comments on the ms.

6. The explanation of exponential models is redundant. In the end you used the log models. Use this notation right from the start and throughout the paper. No need to use exponential models and notation.

7. The data analysis sections must be rewritten. It should follow a logical sequence using subtitles that make sense. This will avoid unnecessary repetition. Controlling overfitting and calculating the most parsimonious models could be subjects for subtitles.

8. The diameter size classes are poorly defined. For instance, a diameter class from 1 to 4 and the next from 4-7 (for instance) should be defined as 1≥ to 4 cm, and the next is 4> to ≤7.

9. The authors compared differences in the averages of some of the morphological characters but did not explain the statistical test used to assess the differences, or whether the data fit the assumptions of the test; they just presented the P value, which precludes the audience from interpreting the power of the test.

10. The expression “was approximately constant” is vague. What does it mean? Is it constant or not? Same for “simple” vs “complex architecture”. These are confusing terms if not elaborated on or defined.

11. The discussion would benefit from a revision of more relevant and pertinent papers. Comparing palms that differ in growth form from the target species is not that relevant since people familiar with palm biology know that palms differ in biomass allocation according to growth form. The discussion should point out the relevance of the data and indicate that calculating so many different models had a utilitarian value.

12. Please respond to the annotations on the ms.

The issues mentioned above could be fixed by the authors. However, the quality of the writing needs to improve in terms of paper structure and organization, the quality of the English language, and the literature revision before the ms is ready for publication. It does make a significant contribution to the estimation of carbon sequestration in peatlands, but the quality of the writing needs to go au pair to the relevance of the data.

Reviewer #2: The authors establish allometric equation for Raphia laurentii a commonest palm in the Central Congo Peatlands. I would like to congratulate the authors for choosing a topic of high importance in the current climate context since Raphia are rarely included in carbon stocks estimation studies due to the absence of specific model to estimate it biomass. However, the manuscript needs to be update before it can be considered for publication. You will see below some my review comment below.

Concerning general comment, it concern the consideration of wood density in the established allometric equation. In fact, like specific species, for me intra-wood density variation cannot be considered like a predicative variable to established R. laurentii biomass but it will be help to address some information in the literature.

Concerning specific comment for each section, see below:

Abstract: Line 23: …… across ca. 45 % of the 23 peatland area. What mean ca?

Line 25-26. Globally in Congo basin, I think it is not totally exclude from AGB estimation but inappropriate methodology approach are used for it biomass estimation (like for example a wrong allometric equation). May be only in the context of Peatland, it is excluded.

L33-34 you mention “ … fitted a linear model relating AGB to each independent predictor variable separately to assess the best variable”. Whereas, we found in L38 a model that taking into account 3 predictive variable; check and correct.

L39’40, when you say about Palm AGB and those of trees estimate at the hectare, it will be interesting if you mention in abstract a synthesis methodology used. Like using establish equation to estimate AGB through palm inventory in 1 plot. And then, how have you make for AGB estimation of trees?

Introduction: I suggest to authors to provide a substantial information on ecological characterization of Peatland with a focus on a R. laurentii description seeing that it represent the principal species composed by peatland.. Again I wanted to know if they exist a general palm allometric equation for Congo basin? If yes, it will be interest to mention it before information provide in the 75-76

Material and methods:

Study area: please can you provide mean altitude of the study area?

L127: remove “in February 2019”

L137-138: why you have not measure the total height directly on the felled R. laurentii?

L224: in abstract you say about 1 ha plot ; here you say about 2 plots. What about?

L230: it will be better to mention better about model using for R. laurentii AGB estimation.

L232: two or one plots?

Result and discussion: Like general comment, R. laurentii is a specific species for which the author gave the wood density. Yes it is a crucial information, for that, the authors can inform the literature about the mean value of WD; however, I don’t know why the author consider the wood density in the allometric equation establish. Personally, I think it is not necessary and I suggest the author to remove all the model that consider WD in this study. For a specific species, intraspecific wood density cannot be considered like predictive variable.

L238: delete “(mean±SE)”;

L238-240: my preoccupation here is if the “SE” is referred to Standard deviation. If it is the case, I am surprised when I see the mean±SE and the variation according to different parameter considered here. In fact, when I take for example the length of fronds, you found 13.30±0.41, that varied from 5.13 to 21.20 m. with that SE 0.41, I am not sure that the variation will be large. This finding is observed from all parameter considered here. In addition, when I see the table 1, you have just transcript in text all information presented here. It will be better in the comment make a like you have make for AGB for example

L266-267. If you conserve 2 number after the comma, make it everywhere

L285: not necessary to consider WD relationship with AGB. The weak relationship can be justified by the fact that it is an intra-specific variation

L294: what is represent x1.01 on the equation provide. I have found the origin or in the model like provide in the table 2

L305-308: yes, Npf is a variable easily to collect in the plot for AGB estimation. However, I think that this model are count between those who over-estimate R. laurentii biomass (according to biais%). Making an over-estimation in it estimation and then a weak accuracy in the final result.

Discussion:

L341-346: be specific by showing that it is due to the fact that it concern different monocot forest trees for which architecture varied. Also,I think that the ecosystem and/or it disturbance can influence also on that proportion.

L385; Table A1?

L391-395: please be clear about what you say. It mean appear to be contrary to information previously found. You mention in L308 that you use the model 16 for R. laurentii estimation; however, here, it appear that you have used all model excepted those that taking into account wood density like predictive variable.

Conclusion: just be focused on the conclusion of your findings. Avoid to discuss again your finding here just conclude.

Tables:

- Table 1. Replace “comma” by “full stop” on different value of wood density

- Table 2. Add a coefficient factor of each model in the table. Add legend of this table; harmonize the number of number after the full stop for each Colum. Moreover, there are some value where it is comma. I suggest also to remove ‘NA’ in the table, and leave these cell empty.

Reviewer #3: Authors should integrate my comments found on the pdf copy, that i deemed necessary

6. PLOS authors have the option to publish the peer review history of their article (what does this mean?). If published, this will include your full peer review and any attached files.

Reviewer #1: **Yes: **Gerardo Avalos

Reviewer #2: **Yes: **Chimi Djomo Cédric

Reviewer #3: **Yes: **Barnabas Neba Nfornkah (Ph.D)

---

## [Author Response · Author response to Decision Letter 0]

2 Dec 2022

Response to Reviewers

We thank the reviewers for their careful assessment of our manuscript, and for describing it as “relevant” (Rev#1) and of “high importance” (Rev#2). The reviewers have made a series of constructive points on the manuscript. We have tried to incorporate as many as possible and think the manuscript greatly improved as a result. We took seriously your comments about the clarity of the text, issues of repetition and the need to improve the quality of the English. As a result, considerable parts of the manuscript have been rewritten and restructured to make the text more coherent and concise, in particular the methods and results sections. The discussion has been rewritten to make it more focused and relevant and to ensure we cite the appropriate literature. 

Response to Referee #1:

Referee #1 General comments: The main goal of this manuscript was to develop allometric equations to estimate AGB for the palm R. laurentii, which dominates the peatlands of the Congo Basin, a critical area for carbon sequestration with the potential to influence global climate. The question is thus relevant, and the authors collected data from 90 individuals. Despite the importance of the question, the manuscript needs to improve significantly in the following aspects:

1. The authors need to significantly improve the use of the English language, the wording, and the paper organization. Many sentences are vague and unclear, have repeated terms, or similar words that follow one another, and the paper has repetitive sections. I tried to mark these issues in the manuscript, but it was a bit too much. The authors need to find help streamlining their writing.

RESPONSE: Please see our response above. We hope the changes we have made have improved the clarity and the readability of the text.

2. The literature review needs to improve. The authors rely on revising allometric equations develop for palms of different growth forms relative to R. laurentii. Some of the comparisons are erroneous in that the compared palms have a significant DBH vs H relationship and R. laurentii is trunkless or has a short stem. Thus, the allometry of the target species is fundamentally different from that of published studies on palms of different growth forms. The discussion should go in the direction of improving the information for palms of different growth habits.

RESPONSE: Thank you for your comments. We have now rewritten large parts of the discussion. We hope it is now clear that we do not wish to make comparisons with other allometric models developed for other palms, but rather to highlight the lack of allomtries suitable for R. laurentii with its different morphology. We have also tried to make better use of the existing literature to help highlight the need for allometries for common species with different morphologies and make the case for the importance of including species such as R. laurentii in regional biomass and carbon stock estimates. 

3. The use of the term “physical parameters” is confusing. Physical could refer to climatic conditions (temperature, radiation, precipitation, etc.) Use instead “morphological variables”.

RESPONSE: We have changed “physical parameters” to “morphological variables” throughout the manuscript.

4. Explain how the 90 palms were selected. Was it a random selection? How spread out were there at the study site? What were the criteria for selection?

RESPONSE: The text now reads “Destructive sampling was carried out approximately 1 kilometres away from the forest plot in an area around 400 m by 500 m. For each mean petiole diameter class we selected 15 individuals at random, each with at least three palm fronds, to be destructively sampled, giving a total of 90 stems.”

5. The figures need to improve. Figure 1´s quality is low. Provide a regional map of the country indicating the study area. Panel b in figure 1 is very dark with a small font size that is hard to read. Figure 2 is unnecessary. The legend of all figures needs to improve and be more accurate. See my comments on the ms.

RESPONSE: We have completely removed the original Figure 2 and have updated the other figures, particularly Figure 1.

6. The explanation of exponential models is redundant. In the end you used the log models. Use this notation right from the start and throughout the paper. No need to use exponential models and notation.

RESPONSE: We now only present the log model in the revised version of the manuscript.

7. The data analysis sections must be rewritten. It should follow a logical sequence using subtitles that make sense. This will avoid unnecessary repetition. Controlling overfitting and calculating the most parsimonious models could be subjects for subtitles.

RESPONSE: The ‘data analysis’ section has been rewritten and restructured to make the methods section clearer and more concise.

8. The diameter size classes are poorly defined. For instance, a diameter class from 1 to 4 and the next from 4-7 (for instance) should be defined as 1≥ to 4 cm, and the next is 4> to ≤7.

RESPONSE: The text now reads: “we chose the following six mean petiole diameter classes for our destructive sampling: 2 to 4 cm, >4 to 5 cm, >5 to 6 cm, >6 to 7 cm, >7 to 8 cm and >8 cm.”

9. The authors compared differences in the averages of some of the morphological characters but did not explain the statistical test used to assess the differences, or whether the data fit the assumptions of the test; they just presented the P value, which precludes the audience from interpreting the power of the test.

RESPONSE: The Kruskal–Wallis test followed by the dunn.test were used to see differences in aboveground biomass, wood density, etc We have added this information to the revised manuscript.

10. The expression “was approximately constant” is vague. What does it mean? Is it constant or not? Same for “simple” vs “complex architecture”. These are confusing terms if not elaborated on or defined.

RESPONSE: The text now reads “There was no significant difference in tissue density between the palm stem and petiole (p>0.05, Dunn’s test; Table 1), but rachis and sheath tissue density were both significantly lower than the stem and petiole tissue density (p value < 0.001, Dunn’s test).” The “simple” and “complex architecture” terms have been removed from the text.

11. The discussion would benefit from a revision of more relevant and pertinent papers. Comparing palms that differ in growth form from the target species is not that relevant since people familiar with palm biology know that palms differ in biomass allocation according to growth form. The discussion should point out the relevance of the data and indicate that calculating so many different models had a utilitarian value.

RESPONSE: We have rewritten large parts of the discussion to make it more relevant to the data we present. We more clearly note that this is the first study of this palm growth form. We do still mention allometries for other palm species as there are no previous studies on allometry of trunkless palms. We add that we calculated a number of different models as they have utilitarian value. 

12. Please respond to the annotations on the ms.

The issues mentioned above could be fixed by the authors. However, the quality of the writing needs to improve in terms of paper structure and organization, the quality of the English language, and the literature revision before the ms is ready for publication. It does make a significant contribution to the estimation of carbon sequestration in peatlands, but the quality of the writing needs to go au pair to the relevance of the data.

Thank you for your comments. We hope that given the substantial changes we have made to the structure and quality of the writing that you find the manuscript greatly improved. 

Response to Referee #2:

Referee #2 General comments: The authors establish allometric equation for Raphia laurentii a commonest palm in the Central Congo Peatlands. I would like to congratulate the authors for choosing a topic of high importance in the current climate context since Raphia are rarely included in carbon stocks estimation studies due to the absence of specific model to estimate it biomass. However, the manuscript needs to be update before it can be considered for publication. You will see below some my review comment below.

Concerning general comment, it concern the consideration of wood density in the established allometric equation. In fact, like specific species, for me intra-wood density variation cannot be considered like a predicative variable to established R. laurentii biomass but it will be help to address some information in the literature.

RESPONSE: We realise that in the original manuscript we did not make it clear as to why we were including wood density (now referred to as “tissue density” in the manuscript, in order to be more accurate) in a species specific allometry. We have added the following text to make clear our rational for including it as a predictor variable: “Although the use of tissue density in a species specific model may initially seem redundant – as all individuals are from the same species with the same mean TD – we have kept this term as it is possible that tissue density may vary depending on environmental conditions. This may be the case give the central Congo peatlands cover 16.7 million hectares, thus R. laurentii are likely to be round growing under different environmental conditions, and may have systematically differing TD. TD can either be assumed to be the same as in this study or measured in situ. Alternatively, we also provide a three parameter model combining SDp, H and Npf provides an almost equally good estimation of R. laurentii AGB, that does not include TD.”

Concerning specific comment for each section, see below: 

Abstract: Line 23: …… across ca. 45 % of the 23 peatland area. What mean ca?

RESPONSE: “ca.” is short for circa, but we have replaced it with “approximately 45% ”.

Line 25-26. Globally in Congo basin, I think it is not totally exclude from AGB estimation but inappropriate methodology approach are used for it biomass estimation (like for example a wrong allometric equation). May be only in the context of Peatland, it is excluded.

RESPONSE: This sentence has been reworded to read “R. laurentii has been excluded from previous aboveground biomass estimates of central Congo peat swamp forest, leading to a systematic bias and underestimation of carbon stocks [5–7].” We hope that this makes it clear that we are referring to AGB estimates for the Congo Basin peatlands specifically.

L33-34 you mention “ … fitted a linear model relating AGB to each independent predictor variable separately to assess the best variable”. Whereas, we found in L38 a model that taking into account 3 predictive variable; check and correct.

RESPONSE: Thank you for pointing out this inconsistency. We have removed this sentence.

L39-40, when you say about Palm AGB and those of trees estimate at the hectare, it will be interesting if you mention in abstract a synthesis methodology used. Like using establish equation to estimate AGB through palm inventory in 1 plot. And then, how have you make for AGB estimation of trees?

RESPONSE: We add in brackets the method for the hardwood trees: “(with hardwood tree AGB estimated using the Chave et al. 2014 allometric equation)”.

Introduction: I suggest to authors to provide a substantial information on ecological characterization of Peatland with a focus on a R. laurentii description seeing that it represent the principal species composed by peatland.. Again I wanted to know if they exist a general palm allometric equation for Congo basin? If yes, it will be interest to mention it before information provide in the 75-76.

RESPONSE: We have rewritten parts of the introduction and have included here a description of Raphia laurentii. We also make it clearer that whilst there have been allometric equations developed for other palm species across the tropics, these are from palms with a different growth form and therefore cannot be applied to Raphia laurentii and as a result, all AGB estimates for the Congo Basin peatlands have excluded Raphia laurentii i.e. have ignored the presence of R. laurentii.

Material and methods:

Study area: please can you provide mean altitude of the study area?

RESPONSE: Done

L127: remove “in February 2019”

RESPONSE: Done.

L137-138: why you have not measure the total height directly on the felled R. laurentii?

RESPONSE: The height is different from the length of frond (from a felled R. laurentii), due to the fronds being very curved and not verticle – so this latter estimate cannot be estimated in the field easily, whereas a height measurement using a hypsometer can be applied in the field, and are trying to develop methods which can be applied easily in the field without destructive sampling. 

L224: in abstract you say about 1 ha plot ; here you say about 2 plots. What about?

RESPONSE: We have now reworded the abstract to make it clear that it was two plots, each 1 ha in side, we applied our allometric equation to.

L230: it will be better to mention better about model using for R. laurentii AGB estimation.

RESPONSE: The methods section has been restructured and we now have a section called “Application of allometric equation” and the this part of the text now reads: “As the only R. laurentii data available for these plots were the Npf for each individual, the Npf single predictor variable linear model for individuals with a mean petiole diameter ≥5 cm (m16, Table 2) was used to estimate R. laurentii AGB.”

L232: two or one plots?

RESPONSE: We have reworded the abstract and the methods and hope it is now clear that it was two 1-hectare plots.

Result and discussion: Like general comment, R. laurentii is a specific species for which the author gave the wood density. Yes it is a crucial information, for that, the authors can inform the literature about the mean value of WD; however, I don’t know why the author consider the wood density in the allometric equation establish. Personally, I think it is not necessary and I suggest the author to remove all the model that consider WD in this study. For a specific species, intraspecific wood density cannot be considered like predictive variable.

&

L285: not necessary to consider WD relationship with AGB. The weak relationship can be justified by the fact that it is an intra-specific variation

RESPONSE: As said above we have now added a section to the discussion to explain our reasoning for including tissue density. We have also added the following text to the discussion when discussing the single predictor variable models: “Unsurprisingly, TD was the only variable which was not suitable for predicting R. laurentii AGB in a single variable model, as it varies little among individuals within the study site.” 

L238: delete “(mean±SE)”;

RESPONSE: Done.

L238-240: my preoccupation here is if the “SE” is referred to Standard deviation. If it is the case, I am surprised when I see the mean±SE and the variation according to different parameter considered here. In fact, when I take for example the length of fronds, you found 13.30±0.41, that varied from 5.13 to 21.20 m. with that SE 0.41, I am not sure that the variation will be large. This finding is observed from all parameter considered here. In addition, when I see the table 1, you have just transcript in text all information presented here. It will be better in the comment make a like you have make for AGB for example.

RESPONSE: We now present the standard deviation instead of the standard error and have added the information which was in this section of the text to Table 1.

L266-267. If you conserve 2 number after the comma, make it everywhere

RESPONSE: Apologies for this error. All numbers are now presented to two decimal places.

L294: what is represent x1.01 on the equation provide. I have found the origin or in the model like provide in the table 2

RESPONSE: This value represents the correction factor (CF=Exp[RSE2/2]), which we have now added to Table 2.

L305-308: yes, Npf is a variable easily to collect in the plot for AGB estimation. However, I think that this model are count between those who over-estimate R. laurentii biomass (according to biais%). Making an over-estimation in it estimation and then a weak accuracy in the final result.

REPONSE: Apologies, we did not make it clear that we used the m16, rather than m5, which underestimates R. laurentii AGB (R2adj = 0.80; Bias (%) = -2.59) considerably less than m5 overestimates AGB. We have now amended the text to make it clearer and it now reads: “When allometric model m16 was applied to the data from the two peat swamp forest plots, the contribution of R. laurentii individuals to the total AGB was 8% and 41% for the hardwood dominated (EKG-02) and R. laurentii dominated (EKG-03) peat swamp plots, respectively (Fig. 6).”

Discussion:

L341-346: be specific by showing that it is due to the fact that it concern different monocot forest trees for which architecture varied. Also,I think that the ecosystem and/or it disturbance can influence also on that proportion.

RESPONSE: We have changed the text to make it clear that these monocots have a very different morphology to Raphia launrentii. The text now reads: “The trunkless morphology of R. laurentii means it is not practical to adopt pre-existing allometries developed for other palm species, which rely on measurements of stem height or stem diameter [9,11,38]. Through our destructive sampling of R. laurentii individuals we show that the palm fronds account for the majority (77%) of their AGB. This is in contrast to similar studies of tree palm AGB for Euterpe precatoria Mart. in the Amazonian Basin [11] and Elaeis guineensis Jacq. in the Congo Basin [38].” With the term “tree palm” definied earlier on in the discussion as “tree palms (i.e. palms with a clearly defined trunk)”.

L385; Table A1?

RESPONSE: Apologies for not spelling it out in full- it referred to Table Appendix 1, however we no longer refer to this table in the text.

L391-395: please be clear about what you say. It mean appear to be contrary to information previously found. You mention in L308 that you use the model 16 for R. laurentii estimation; however, here, it appear that you have used all model excepted those that taking into account wood density like predictive variable.

RESPONSE: We have rewritten this section of the discussion and now clearly say what we think is the best three predictor variable mode and the best single predictor variable model in terms of model performance and the easiest model to apply in the field.

Conclusion: just be focused on the conclusion of your findings. Avoid to discuss again your finding here just conclude.

RESPONSE : We have rewritten the conclusions to ensure it summarises what has already been said in the paper without introducing any new topic or interpretation of the data. 

Tables:

Table 1. Replace “comma” by “full stop” on different value of wood density

RESPONSE: Apologies. Done.

Table 2. Add a coefficient factor of each model in the table. Add legend of this table; harmonize the number of number after the full stop for each Colum. Moreover, there are some value where it is comma. I suggest also to remove ‘NA’ in the table, and leave these cell empty.

RESPONSE: The table has been corrected for inconsistencies in number formatting and a more detailed legend has been provided. Correction factors have also been added to table 2.

Response to Referee #3:

Referee #3 General comments: Authors should integrate my comments found on the pdf copy, that i deemed necessary

L.1. "... equation for Raphia laurentii, the commonest palm..."

RESPONSE: The title now reads “Allometric equation for the commonest palm in the Central Congo Peatlands, Raphia laurentii De Wild.” 

L.22. Raphia... is the most abundant palm in this peatland area..." More effective to start with the target species right away

RESPONSE: The text now reads “Raphia laurentii De Wild., the most abundant palm in these peatlands, forms dominant to mono-dominant stands across approximately 45% of the peatland area”

L.24. provide a brief description of the palm once you name it instead of mixing this description with the objective of the study

RESPONSE: The text now reads “R. laurentii is a trunkless palm with fronds up to 20 m long that can be canopy-forming. Owing to its morphology, there is currently no allometric equation which can be applied to R. laurentii. Therefore it is currently excluded from aboveground biomass (AGB) estimates for the Congo Basin peatlands.”

L.27. confusing to explain size classes here where space is limited. Do it in Methods. At any rate, you used "mean size classes"... what does it mean? Classes are usually defined across a range. This is why this level of detail needs to go in Methodds

RESPONSE: We have now removed this from the abstract.

L.29. "...sampling, we measured diameter at..." drop the :

RESPONSE: Text now reads “Prior to destructive sampling, stem base diameter, petiole mean diameter, the sum of petiole diameters, total palm height, and number of palm fronds were measured.”

L.30. you mean the diameter of the petiole of the actual width of the frond?

RESPONSE: Apologies for the lack of clarity. We meant petiole. The text now reads: “Prior to destructive sampling, stem base diameter, petiole mean diameter, the sum of petiole diameters, total palm height, and number of palm fronds were measured.”

L.31. height, insert comma

RESPONSE: Done.

L.32-33. this part should go first... you divided the palms into these modules, and then weight them... follow a logical sequence

RESPONSE: Text now reads “After destructive sampling, each individual was separated into stem, sheath, petiole, rachis, and leaflet categories, then dried and weighed.”

L.34. and the predictor variables were?

RESPONSE: Owing to rewording of the abstract, this part of the text has been removed.

L.34. the model with the best fit

RESPONSE: Owing to rewording of the abstract, this part of the text has been removed.

L.35. you mentioned "petiole diameter classes" above... not clear if you are referring to this, or to stem diameter, and if so, how many classes were defined?

RESPONSE: To simplify the text we have removed reference to the diameter classes in the abstract.

L.38. missing a parenthesis to close the equation after the Exp

RESPONSE: Now added.

L.40. are you referring to palms as "trees"? Are there were actual trees in the plots? This sentence is confusing

RESPONSE: Text now reads “We applied one of our allometric equations to data from two nearby 1-hectare forest plots, one dominated by R. laurentii, where R. laurentii accounted for 41% of the total forest AGB (with hardwood tree AGB estimated using the Chave et al. 2014 allometric equation), and one dominated by hardwood species, where R. laurentii accounted for 8% of total AGB.”

L.40-41. the ending sentence is weak. You are referring to this spp in particular, not to all palms. Considering that peatlands in Congo are a very important reservoir of C, and that now this area is threaten by development, you have a more compelling case by referring to the important of these habitats within the context of climate change. Thus, make a better, more compelling case.

RESPONSE: We have rewritten the end of the abstract, which now finishes with:“Across the entire region we estimate that R. laurentii stores around 2 million tonnes of carbon aboveground. The inclusion of R. laurentii in AGB estimates, will drastically improve overall AGB, and therefore carbon stock estimates for the Congo Basin peatlands.”

L48-49: avoid using similar words in the same sentence

RESPONSE: We have now substituted the word “estimating” with “quantifying”

L55: "... pools (i.e., coarse woody debris...)". drop the reference to monocotyledonous and dicotyledonous vegetation. It is obvious

RESPONSE: As the other carbon pools are not directly relevant to this study we have simplified this sentence to read: “However, information on reference levels requires accurate estimates of the biomass, and therefore the carbon stock, of the different forest types”

L57: Establishing carbon reference levels in the Congo Basin is hindered by the lack of data.

RESPONSE: We have replaced “a scarcity of data” with “a lack of data”.

L60: who is the subject here? Need to improve writing to make it less vague

RESPONSE: This sentence has been removed.

L60: you are mixing 2 things here... a. carbon estimates use allometric equations developed for trees, not palms. b. These estimates were developed in terra firme forests, not peatlands.

RESPONSE: We agree. We have removed this sentence as it is not relevant to the lack of data on palm biomass.

L62: what´s the difference between a dominant and a monodominant stand? It is the same

RESPONSE: A dominant stand is where the species is the most abundant either in terms of number or biomass, whereas a monodominant stand is where the species is the sole canopy species present in that stand.

L63: bad word choice. What is an "obvious" equation? The word is vague when applied to equations

RESPONSE: The word “obvious” has been removed.

L64-65: Palms have been excluded from previous estimates of... Put the subject at the beginning of the sentence... sentences must be active.

RESPONSE: Now reads: “R. laurentii has been excluded from previous aboveground biomass estimates of central Congo peat swamp forest, leading to a systematic bias and underestimation of carbon stocks [5–7].”

L66: subestimation of C sequestration

RESPONSE: We have kept the word “underestimation” as it is the more commonly used term.

L70-73: improve the writing... you repeat "allometric equation" several times

RESPONSE: This section now reads: “Whilst some allometric equations have been developed for Amazonian [9,11] and Asian [12,13] palm species, these palms are morphologically dissimilar to R. laurentii. Generally, in monocot species, using a species-specific model is better than using a multispecies model [9].” 

L80: information to inform is redundant. 

RESPONSE: Now reads: “Without this information to guide”

L81: what sort of development?. This should be a separate sentence from the previous one

RESPONSE: There are a number of potential developments which threaten the peatlands discussed in the paper we cite.

L82: to morphological variables

RESPONSE: Now reads: “to morphological variables”

L87-88: the importance of a given character depends on the spp, but you are saying these are the most important in descending order. Where is the reference? In my experience, this statement is inaccurate.

RESPONSE: We now cite the following two studies to support this statement:

Chave J, Andalo C, Brown S, Cairns MA, Chambers JQ, Eamus D, et al. Tree allometry and improved estimation of carbon stocks and balance in tropical forests. Oecologia. 2005;145: 87–99. doi:10.1007/s00442-005-0100-x

Fayolle A, Ngomanda A, Mbasi M, Barbier N, Bocko Y, Boyemba F, et al. A regional allometry for the Congo basin forests based on the largest ever destructive sampling. For Ecol Manag. 2018;430: 228–240. doi:10.1016/j.foreco.2018.07.030.

L89: In contrast,

RESEPONSE: Now reads “in contrast”

L90: predictor variables

RESPONSE: Now reads “predictor variables”

L92: it is not unclear. Some palms have a DBH vs H relationship, some others don´t, especially the ones in which the DBH vs H increase is decoupled. See for instance Avalos, G., Gei, M., Ríos, L. D., Otárola, M. F., Cambronero, M., Alvarez-Vergnani, C., ... & Rojas, G. (2019). Scaling of stem diameter and height allometry in 14 neotropical palm species of different forest strata. Oecologia, 190(4), 757-767. Renninger, H. J., & Phillips, N. (2010). Intrinsic and extrinsic hydraulic factors in varying sizes of two Amazonian palm species (Iriartea deltoidea and Mauritia flexuosa) differing in development and growing environment. American journal of botany, 97(12), 1926-1936.

RESPONSE: We think the reviewer may have misunderstood what we have said. We are referring to trunkless palms when we say it is unclear. To improve clarity we have reworded the text: “For palms in particular, height and diameter (at the base or at breast height) are the most important morphological variables for estimating above-ground biomass [9,12], but for palms with morphologies like R. laurentii, other predictor morphological variables may be more practical and therefore preferential.”

L95: this should go first in the paragraph

RESPONSE: Now reads “The aim of this study is to develop an allometric equation for the monocotyledonous species R. laurentii, in order to improve the aboveground biomass and carbon stock estimates of the peat swamp forests in the Congo Basin.”

L.105. are, respectively,

RESPONSE: Done.

L118-119: obvious and redundant... you said this already

RESPONSE: Sentence removed.

L. 120: how tall?

RESPONSE: Now reads: “with 4 to 6 stems (i.e. short trunks) which grow to 2 – 7 m tall and up to 20 cm in diameter [8].” and this description has been moved to the introduction.

L122: ???

RESPONSE: Word “evolution” replaced by “growth”.

L123-124: so the stems are higher than 1.3m?

RESPONSE: See added text above in response to L120 query.

L.126: how did you select these subplots? location?

RESPONSE: We have edited this section to read “In order to construct a robust allometric equation which could be applied to the full size-range of R. laurentii, palm frond diameter data were collected prior to destructive sampling from a nearby 1 ha forest plot (EKG-03; 1.191998N, 17.84693E), located in a R. laurentii dominated forest.”

L132: it is important to indicate 2 to >= 4, >4 to <=5... I don´t have the correct sign for <= or >= here but do you see my point? Does the range includes 4? Because if it does the next range cannot include 4, it should be >4...

RESPONSE: Text now reads: “we chose the following six mean petiole diameter classes for our destructive sampling: 2 to 4 cm, >4 to 5 cm, >5 to 6 cm, >6 to 7 cm, >7 to 8 cm and >8 cm.”

L134: how many individual palms did you sample? This needs to be indicated early in the paragraph

RESPONSE: Text now reads: “For each mean petiole diameter class we selected 15 individuals at random, each with at least three palm fronds, to be destructively sampled, giving a total of 90 stems.”

L.135: is this relevant for the allometric equation? I don´t think so. Explain the data that is relevant for the objectives. Is location going to be analyzed? Nope

RESPONSE: Text removed.

L.139: ?? in response to Vertex

REPONSE: Text now reads: “using a hypsometer (manufacturer: Haglöf Sweden, Långsele, Sweden; model: Vertex IV)”

L.139: not clear what you measured here... the stem diameter at 1.3m or the petiole diameter?

RESPONSE: The text now reads “The diameter of each petiole was measured”

L.140: this needs to be explained better... the point of measurement of stem diameter was at the emergence of the petioles? the lowest one?

RESPONSE: Text now reads: “The diameter of each petiole was measured using callipers at 1.3 m from the ground or, depending on the point of petiole emergence, above 1.3 m, whilst avoiding the sheath.”

L.160. here it is relevant to indicate the capacity and model of this balance 

RESPONSE: Have added information on the manufacturer, model and capacity.

L.166. model?

RESPONSE: Have added information on the manufacturer, model and capacity.

L. 171. model?

RESPONSE: Have added information on the manufacturer, model and capacity.

L.171-172. some details of the process are irrelevant. Obviously you corrected for the weight of the plastic bag.

RESPONSE: This text has been removed.

L.173. volume difference

RESPONSE: We believe hydrostatic weighing to be the correct term.

L. 174. 101 is too hot... it is likely that you lost some biomass specially parenchyma cells that turned to ashes inside the tissues. Do you have a reference justifying this temp for drying biomass?

REPONSE: We were fllowing the protocol of Goodman et al. 2013 and now include that reference.

L,176. across all modules? What modules did you measure for this? stem, petiole?

REPONSE: The text now reads: “The tissue density (TD) of each sample (stem, sheath, petiole and rachis) was calculated as dry mass divided by volume [27].”

L. 183. evaluating... what? Check the grammar

RESPONSE: The heading now reads: “Development of allometric equations”

L.191. obviously the morphological variables are not independent. For instance, if you use stem diameter, height, number of fronds, etc.. all are correlated. However, there is a functional relationship that makes the argument to define them as predictor variables... the hypothesis could be that diameter is related to mechanical support which is associated to the size or biomass of the palm... etc. and so the argument goes. But assuming that the morphological characters or an individual are independent be default is wrong.

RESPONSE: Apologies for the miswording of this. We now refer to them as “predictor variables” throughout the text. 

L.191. intersection with Y axis

RESPONSE: Text now reads: “a is the intercept”.

L191. exponent in this case, but slope values associated with the predictor variables in a log equation

RESPONSE: Text now reads: “b is the slope”.

L.193. does that corrected the lack of normality? Did you check?

RESPONSE: Yes we checked it with a Kurtosis test in the R package “gvlma” (results shown below). To make it clearer that the transformation resolved the issue we have modified the text slightly to read: “To correct for heteroscedasticity, the predictor variables were log transformed.”

L.194. using the first equation is right... then you said you used the log model... which is also right. However, presenting the first equation is redundant if you just used the second one

RESPONSE: We now only present the log model.

L.197. see Sprugel, D. G. (1983). Correcting for bias log-transformed allometric equations. Ecology 64, 209–210. doi: 10.2307/1937343

RESPONSE: We now cite Sprugel (1983).

L.207. below you are discussing akaike values...I suggest you reorganize this section and follow a logical sequence since you are explaining model selection here... then below explain the fit and akaike in a separate section... it is a bit messy 

RESPONSE: The methods section has been substantially restructured and slimmed down. We hope this has improved the clarity of this section.

L.208. consider the akaike criterion. Usually this is consistent with RSE... Include akaike for comparative purposes with other studies

RESPONSE: We think that R2 and RES are reseaonable to compare models which have only one predictor variable. In our knowledge, Adjusted R2, AIC, RSE, RMSE and Bias are often used to compare models with more than one predictor variable.

L. 215. should explain if this protocol resulted in the best parsimonious models, not clear from the writing. It needs to compare the fit (R2) with the number of variables in the model

RESPONSE: The text now reads: “The overall best performing model was the one with the highest R2adj value, the smallest AIC, RSE, Roots Mean Square Error (RMSE) and Bias prediction (Bias%=1/n ∑_(i=1)^n▒[〖(AGB〗_est-〖AGB〗_observed)/〖AGB〗_observed ] ) values [33]. A partial Fisher’s test was used to test the validity of including the second and third predictor variables.”

L.217. data analysis needs to be separated in sections. What you did above is also part of "Data Analysis". Use more specific subtitles

RESPONSE: We have restructured the methods and now all the fitting and analysis of model performance is detailed under a section called “Development of allometric equations”.

L.219. don´t need to include the formula

RESPONSE: This has been removed.

L.222. ok... you need to organize your data analysis better to avoid being repetitive

RESPONSE: We have restructured the methods and results considerably to remove repetition.

L.224. should be a separate section... Model validation???

RESPONSE: Please see our comment above in response to L.217.

L.226-227. it is not a complete sentence

RESPONSE: This sentence now reads: “Within these two plots the AGB of every tree with a DBH ≥10 cm was estimated using the allometric equation of Chave et al. [35].”

L.228. if

RESPONSE: In the rewording of this text, this sentence no longer exists.

L.230. eliminate the .

RESPONSE: In the rewording of this text, this sentence no longer exists.

L.230. which one? Number your models

RESPONSE: Now numbered. 

L.232. again the English fails a bit... please indicate how the comparison was done... the estimate with your model and the estimate w/o your model.

RESPONSE: This section has been reworded and now reads as follows: “To assess the impact of the inclusion of R. laurentii on AGB estimates for the Congo Basin peatlands, we took data from two 1-ha forest peatland plots and compared the AGB estimates when palms were included and excluded. The two plots from which the data derived were the R. laurentii dominated plot (EKG_03), used to determine the mean petiole diameter classes, and a hardwood dominated swamp plot (EKG_02; 1.19200, 17.84693). Within these two plots the AGB of every tree with a DBH ≥10 cm was estimated using the allometric equation of Chave et al. [35]. As the only R. laurentii data available for these plots were the Npf for each individual, the Npf single predictor variable linear model for individuals with a mean petiole diameter ≥5 cm (m16, Table 2) was used to estimate R. laurentii AGB. We then compared the AGB estimates of the two plots when only tree AGB was included (as has previously been the practice [5]) and when both R. laurentii and tree AGB was included.” 

L232. Obviously there is going to be a big difference b/c in one case you are adding palms that are very abundant. A more valid comparison is to estimate palm biomass using Chave or a general palm model, different from your specific model. There are several palm models in the literature

RESPONSE: This is obviously a big difference. But we cannot use another palm model, as they are from a very different mortphology, as R. laurentii has a trunkless form. The reason for including this comparison, however, was to highlight that without the means to estimate R. laurentii AGB, the peatland AGB is currently considerably underestimated, even in swamp forest where it is not dominant.

L.234. at the end indicate that you used R software and name the different libraries that you consulted and applied

RESPONSE: The following has now been added to the text: “All statistical analyses were carried with R (http://www.r-project.org/) [28], using the packages “ggplot2” [29], “minpack.lm” [30], “car” [31], “dunn.test” [32], “gvlma” [33] and “ggpubr” [34].”

L.238-243. could it be possible to present these data in a Table?

RESPONSE: Now presented in Table 1.

L.246. this goes in Discussion... btw, where is the reference?

RESPONSE: This has been removed.

L.247. a test here is necessary

RESPONSE: The text now reads “There was no significant difference in tissue density between the palm stem and petiole (p>0.05, Dunn’s test; Table 1), but rachis and sheath tissue density were both significantly lower than the stem and petiole tissue density (p value < 0.001, Dunn’s test).”

L.249. what is the test that was applied here?

RESPONSE: The text now reads “The dry mass/fresh mass ratio was significantly higher (p value < 0.001, Dunn’s test) in the palm frond compartments, compared to the stem (Table 1).”

L.257. biomass partitioning.

RESPONSE: As table 1 now also includes data on density and dry mass/fresh mass ratios the legend now reads: “Table 1. Physical characteristics of the 90 R. laurentii individuals destructively sampled in the peat swamp forest, Likouala Department, Republic of the Congo.”

L.257. use SD instead. SD is the most common value reported in the lit

RESPONSE: Done.

L.262. ???

RESPONSE: Reworded to read: “Mean palm frond AGB (44.93±50.87 kg) was significantly higher than mean stem AGB (11.26±19.49 kg; p < 0.001, Kruskal-Wallis test; Fig. 2a). AGB increased with increasing mean petiole diameter class (Fig. 2b).The allocation of biomass between each component (i.e., stem, sheath, petiole, rachis and leaflets) was did not vary greatly between mean petiole diameter classes (Fig. 3).”

L.273. the actual averages are very close.

RESPONSE: We are considering the sum of all palm frond compartments, which are is higher than the stem AGB.

L.274. give the value of the test with the sample size

RESPONSE: Reworded to read: “Mean palm frond AGB (44.93±50.87 kg) was significantly higher than mean stem AGB (11.26±19.49 kg; p < 0.001, Kruskal-Wallis test; Fig. 2a).”

L.276. the legend is incomplete. Instead of compartments refer to fronds and stems... just 2 variables... name them.

RESPONSE: Now reads: “Fig 2. Variability of aboveground biomass between stem and palm fronds (a) across mean petiole diameter classes (b) for the 90 Raphia laurentii individuals destructively sampled.”

L.276. the legend should help the audience to interpret the figure w/o having to go to the main text

RESPONSE: Please see directly above.

L.279. defective wording gets in the way of understanding the science. This is one example. You must seek help from a native speaker with some experience in this field to streamline the writing

RESPONSE: We have reworded this section considerably and hope it now reads better.

L.282. you should make reference to the table showing the models instead... there the audience could see models with one predictor variable and assess their fit

RESPONSE: Done.

L.285-286. this is not a very effective way of presenting the data. Indicate the relevant equations from the table and then make reference to the table...

"The most parsimonious model was XXX... (Table 2)".

RESPONSE: We have rewritten this section to make it clearer, making reference to specific models and Table 2. We have also highlighted the best performing models in Table 2 to help the reader.

L.289. The writing must improve. Indicate what models had a good fit by referring to the table, model #, name of variable and R2. Pinpoint only the important trends.

RESPONSE: Please see above.

L.294. present the model in log form... this switching back and forth b/w exponential and logarithmic is confusing and redundant. Use one or the other.

RESPONSE: Done.

L.307. what is the R2 value here... in this instance it is important to justify this choice.

RESPONSE: We apologise for not being clear about our reasoning for selecting this model. To make it clear we have added the following to the methods: “As the only R. laurentii data available for these plots were the Npf for each individual, the Npf single predictor variable linear model for individuals with a mean petiole diameter ≥5 cm (m16, Table 2) was used to estimate R. laurentii AGB.”

Table 2: 

-equations

-highlight the most parsimonious model... highlight the model you recommend

- where is the correction factor?

- is there a model across all diam ranges?

RESPONSE: We have now added the correction factor to table 2 and highlighted the best performing models. The first section of the Table 2, where it says the mean petiole diameter class is 2-15 cm (n=90), is for all the diameter ranges.

L.347. this discussion is irrelevant. It is clear that some palms have a significant DBH vs H relationship and thus allocate most of their biomass in the stem... other species have a different growth strategy... they first develop sufficient diameter at the base before increasing in height. Then, there are palms that are trunkless, like in this case. Thus, making the comparison at this level is not very informative. The discussion should be going in the direction of adding data to quantify carbon sequestration across the diversity of growth forms found in palms. See the following:

https://www.frontiersin.org/articles/10.3389/ffgc.2022.1021784/full

and articles therein

RESPONSE: We have rewritten the discussion. We use the discussion to highlight the need for allometries for palm with morphologies that differ from tree palms and the relevance of our Raphia laurentii allometries when quantifying aboveground carbon stocks for the Congo Basin peatlands. We have included new citations as suggested.

L.348. what do you mean by "simple architecture"?

RESPONSE: We have now removed this term. We have tried to make it clear throughout the manuscript that pre-existing palm allometries only exist for palms with a clear, visible, measureable stem and that R. launrentii, whilst it technically has a stem, does not have a stem which is visible or accessible without destructive sampling. For example we have added “We refer to R. laurentii as a trunkless palm, despite the presence of a stem in mature individuals, because their growth in compact clumps and the high number of fronds per clump, means the stem is not accessible and rarely visible. This makes it difficult to measure the diameter at the base of the stem or at 1.30 m from the ground, without destructive sampling.” [L65-69 in the new manuscript version] and “tree palms (i.e. palms with a clearly defined trunk)” [L306-307 in the new version of the manuscript].

L.351. again, what do you mean by "complex"? The reference you cite doesn´t explain it

RESPONSE: This term has been removed. Please see directly above for a fuller explanation of the changes we have made in response to this comment.

L.354-355. this happens b/c the stem is small... it is the growth form that matters.

RESPONSE: We have reworded the text to make it clearer that the morphology of R. laurentii differs from other palms and therefore the key predictor variable for AGB differ: “Here however, the concentration of AGB in the palm fronds, meant that stem diameter and total palm height were the some of the worst performing single variable predictors of R. laurentii AGB (Table 2).”

L.355. another term that is vague... "Practical allometry"?

Reading between the lines I guess you meant "easier to measure in the field to estimate AGB". 

RESPONSE: Because we have restructured and rewritten parts of the discussion this sentence no longer exists. But we do discuss the practical advantages of our allometries in the discussion, particularly in the section “Allometric equation for AGB estimation”.

L.360. and also to the fact that Raphia has the more massive leaves of all Arecaceae if not of all the Plant Kingdom

RESPONSE: Because we have restructured and rewritten parts of the discussion this sentence no longer exists.

L.382. The discussion should go in the direction that there are potential users of these models out there... then, you have to provide options with variables that these users could use to estimate AGB... not only one model

RESPONSE: In the “Allometric equation for AGB estimation” we discuss the different models and their advantages in terms of performance but also their practical applicability in the field, so potential users can understand the different options we present.

L.401. how much biomass is below ground? 

RESPONSE: In this study we only considered the aboveground biomass.

Fig. 1. the second panel MUST improve. It is almost invisible. The labels are tiny, the color quality bad. Show the location at the regional level.

RESPONSE: We have amended Figure 1 and hope it is now of an acceptable standard. 

Fig. 2. this is petiole diameter. I don´t think this figure is necessary. Explaining how the data was taken in the text is sufficient. It uses space unnecessarily 

RESPONSE: This has been removed.

Fig. 3. Consider using box plots per compartment to show the variation among the 90 individual palms

RESPONSE: Instead we have modified the former Figure 5 (now Figure 4) to show boxplots of the different compartments.

Fig. 4. figures need to indicate what statistical tests were done for significance. Figures need to be interpreted without having to recur to the main text. 

second set of box plots tiny...

only fronds and stem

it is mean petiole diameter. Please use accurate labels

RESPONSE: We now include the statistical test and results used and have updated the legend to read: Fig 2. Variability of aboveground biomass between leaflets, petiolem, rachis, sheath and stem, and the results of a Kruskal-Wallis test (a) and between palm frond compartments collectively and stem across mean petiole diameter classes (b) for the 90 Raphia laurentii individuals destructively sampled.

Fig. 6. goodness of fit

RESPONSE: Legend now reads: “Fig 5. Mean relative error for best performing aboveground biomass (AGB) models for R. laurentii individuals with mean palm petiole diameters of ≥2 cm (model 11) and ≥5 cm (model 18) across different AGB ranges.”

Fig. 7. forest type is unnecessary. The legend needs to improve. The third bar is unnecessary

RESPONSE: This figure has been updated accordingly.

---

## [Decision Letter · Decision Letter 1]

27 Feb 2023

Allometric equation for Raphia laurentii De Wild, the commonest palm in the Central Congo Peatlands.

PONE-D-22-22408R1

Dear Dr. Bocko,

We’re pleased to inform you that your manuscript has been judged scientifically suitable for publication and will be formally accepted for publication once it meets all outstanding technical requirements.

Kind regards,

John Toland Van Stan II, Ph.D.

Academic Editor

PLOS ONE

Additional Editor Comments (optional):

Thank you for revising your manuscript per the suggestions of three reviewers. Two reviewers have recommended acceptance. The remaining reviewer suggests rejection; however, I find that this reviewer's criticism does not align with the revised manuscript. Specifically, they cite concerns regarding the lack of important revisions that they requested on figures and tables, but I can clearly see these revisions in the new manuscript. I emailed the reviewer to see if I was missing something, or if they had simply reviewed the wrong materials. After a few weeks, the reviewer has not responded. I have read the revised manuscript and the author response letter again. My opinion is that the revisions were major, that they addressed the reviewer concerns, and that the paper now meets the publication criteria of PLOS one. Therefore, I recommend its acceptance. Congratulations!

Reviewers' comments:

Reviewer's Responses to Questions

**Comments to the Author**

1. If the authors have adequately addressed your comments raised in a previous round of review and you feel that this manuscript is now acceptable for publication, you may indicate that here to bypass the “Comments to the Author” section, enter your conflict of interest statement in the “Confidential to Editor” section, and submit your "Accept" recommendation.

Reviewer #1: (No Response)

Reviewer #2: All comments have been addressed

2. Is the manuscript technically sound, and do the data support the conclusions?

Reviewer #1: No

Reviewer #2: Yes

3. Has the statistical analysis been performed appropriately and rigorously? 

Reviewer #1: No

Reviewer #2: Yes

4. Have the authors made all data underlying the findings in their manuscript fully available?

Reviewer #1: Yes

Reviewer #2: Yes

5. Is the manuscript presented in an intelligible fashion and written in standard English?

Reviewer #1: No

Reviewer #2: Yes

6. Review Comments to the Author

Reviewer #1: I was surprised to see that the authors basically did very superficial revisions and ignored the majority of the comments made in the first revision. Thus, the same issues noted in the first revision still remain. For instance, figure 1 is still very dark and with little detail. It does not give a clear idea of the location and geographic context of the study site. Figure 2 works ofr a powerpoint presentation, not for a paper. The same problems of writing and manuscript organization pointed out in the first manuscript persisted. For example, the first part of the introduction is marginal to the study since comparisons with species showing clear differences in growth forms is irrelevant. The discussion should have focused, instead, on allometric models. Table 1 does not highlight the recommended and most parsimonious models. Ambiguous expressions such as "simple vs complex architecture", etc. still persist. I could submit my corrections from the first revision and thet would still apply to this second revision. Since the authors did a cursory review and did not significantly improve the manuscript, my recommendation is to reject it. Although the data and context is valuable, the writing does not have the level necessary to be published.

Reviewer #2: (No Response)

7. PLOS authors have the option to publish the peer review history of their article (what does this mean?). If published, this will include your full peer review and any attached files.

Reviewer #1: No

Reviewer #2: **Yes: **Chimi Djomo Cédric

---

## [Editor Report · Acceptance letter]

5 Apr 2023

PONE-D-22-22408R1 

Allometric equation for *Raphia laurentii* De Wild, the commonest palm in the Central Congo Peatlands. 

Dear Dr. Bocko:

I'm pleased to inform you that your manuscript has been deemed suitable for publication in PLOS ONE. Congratulations! Your manuscript is now with our production department. 

Kind regards, 

on behalf of

Dr. John Toland Van Stan II 

Academic Editor

PLOS ONE